



# Lower thermosphere - ionosphere (LTI) quantities: Current status of measuring techniques and models

Minna Palmroth[1,2], Maxime Grandin[1], Theodoros Sarris[3], Eelco Doornbos[4], Stelios Tourgaidis[3,5], Anita Aikio[6], Stephan Buchert[7], Mark A. Clilverd[8], Iannis Dandouras[9], Roderick Heelis[10], Alex Hoffmann[11], Nickolay Ivchenko[12], Guram Kervalishvili[13], David J. Knudsen[14], Anna Kotova[9], Han-Li Liu[15], David M. Malaspina[16,17], Günther March[18], Aurélie Marchaudon[9], Octav Marghitu[19], Tomoko Matsuo[20], Wojciech J. Miloch[21], Therese Moretto-Jørgensen[22], Dimitris Mpaloukidis[3], Nils Olsen[23], Konstantinos Papadakis[1], Robert Pfaff[24], Panagiotis Pirnaris[3], Christian Siemes[18], Claudia Stolle[13,25], Jonas Suni[1], Jose van den IJssel[18], Pekka T. Verronen[2,26], Pieter Visser[18], and Masatoshi Yamauchi[27]

[1]University of Helsinki, Department of Physics, Helsinki, Finland
[2]Space and Earth Observation Centre, Finnish Meteorological Institute, Helsinki, Finland
[3]Department of Electrical and Computer Engineering, Democritus University of Thrace, Xanthi, Greece
[4]Royal Netherlands Meteorological Institute KNMI, Utrecht, The Netherlands
[5]Space Programmes Unit, Athena Research & Innovation Centre, Athens, Greece
[6]Space Physics and Astronomy Research Unit, University of Oulu, Oulu, Finland
[7]Swedish Institute of Space Physics (IRF), Uppsala, Sweden
[8]British Antarctic Survey (UKRI-NERC), Cambridge, UK
[9]Institut de Recherche en Astrophysique et Planétologie, Université de Toulouse, CNRS, CNES, Toulouse, France
[10]Center for Space Sciences, University of Texas at Dallas, Dallas, USA
[11]European Space Research and Technology Centre, European Space Agency, Noordwijk, The Netherlands
[12]Royal Institute of Technology KTH, Stockholm, Sweden
[13]GFZ Potsdam, German Research Centre for Geosciences, Potsdam, Germany
[14]Department of Physics and Astronomy, University of Calgary, Calgary, Canada
[15]National Center for Atmospheric Research, Boulder, Boulder, USA
[16]Astrophysical and Planetary Sciences Department, University of Colorado, Boulder, USA
[17]Laboratory for Atmospheric and Space Physics, University of Colorado, Boulder, USA
[18]Faculty of Aerospace Engineering, Delft University of Technology, Delft, The Netherlands
[19]Institute for Space Sciences, Bucharest, Romania
[20]Ann and H.J. Smead Department of Aerospace Engineering Sciences, University of Colorado at Boulder, Boulder, USA
[21]Department of Physics, University of Oslo, Oslo, Norway
[22]University of Bergen, Institute of Physics and Technology, Bergen, Norway
[23]DTU Space – Technical University of Denmark, Copenhagen, Denmark
[24]Heliophysics Science Division, NASA/Goddard Space Flight Center, Greenbelt, USA
[25]University of Potsdam, Faculty of Science, Potsdam, Germany
[26]Sodankylä Geophysical Observatory, University of Oulu, Sodankylä, Finland
[27]Swedish Institute of Space Physics (IRF), Kiruna, Sweden

**Correspondence:** Minna Palmroth (minna.palmroth@helsinki.fi)

**Abstract.**





The lower-thermosphere–ionosphere (LTI) system consists of the upper atmosphere and the lower part of the ionosphere, and as such comprises a complex system coupled to both the atmosphere below and space above. The atmospheric part of the LTI is dominated by laws of continuum fluid dynamics and chemistry, while the ionosphere is a plasma system controlled by electromagnetic forces driven by the magnetosphere and solar wind. The LTI is hence a domain controlled by many different physical processes. However, systematic *in situ* measurements within this region are severely lacking, although the LTI is located only 80 to 200 km above the surface of our planet. This paper reviews the current state of the art in measuring the LTI, either directly or by several different remote-sensing methods. We begin by outlining the open questions within the LTI requiring high-quality *in situ* measurements, before reviewing directly observable parameters and their most important derivatives. The motivation for this review has arisen from the recent retention of the Daedalus mission as one among three competing mission candidates within the European Space Agency (ESA) Earth Explorer 10 Programme. However, this paper intends to cover the LTI parameters such that it can be used as a background scientific reference for any mission targeting *in situ* observations of the LTI.

## 1  Introduction

The region where the atmosphere meets space, called the Lower Thermosphere–Ionosphere (LTI), is sometimes also termed as the *ignorosphere* because it is markedly difficult to make measurements of it directly. This region, spanning from about 80 km to 200 km in altitude, exhibits a relatively high atmospheric density, making systematic satellite *in situ* measurements impossible from circular orbits. This is the region where de-orbiting spacecraft and orbital debris start to burn up while re-entering the atmosphere. Hence sporadic rocket campaigns are currently the main source of *in situ* observations (e.g., Burrage et al., 1993; Brattli et al., 2009). Remote optical observations require measurable emissions reaching the remote detector; however, there is a significant gap in ultraviolet, infrared and optical emissions (for Fabry-Perot interferometers) at approximately 100–140 km altitude, allowing only a part of the LTI to be measured remotely. Ground-based radar measurements are also inherently remote, but are indispensable especially in characterising the ionised part of the LTI, the ionosphere. Due to the lack of systematic measurements, this region still yields discoveries and surprises; for instance, as recently reported by Palmroth et al. (2020), even Citizen Scientist pictures of the aurora may be relevant in obtaining new information on the LTI.

A few comprehensive reviews of the LTI have been published in the recent years. Vincent (2015) concentrates in the atmospheric dynamics within the region. Laštovička (2013) and Laštovička et al. (2014) review the trends in the observational state-of-the-art within the upper atmosphere and ionosphere. Sarris (2019) reviews the characterisation status and presents the key open questions especially in terms of measurement gaps within the LTI, while also highlighting the discrepancies between observations and models. A recently accepted review article by Heelis and Maute (2020) describes the challenges within the understanding of the LTI in terms of coupling to the lower atmosphere, the LTI as a source of currents, its coupling to regions



above, and the response of the LTI to different drivers. Apart from these recent reviews, one of the most thorough introductions to the LTI dates back to a 1995 book within the American Geophysical Union Geophysical Monograph Series, reviewing,

among other aspects, the dynamics of the lower thermosphere (Fuller-Rowell, 2013). These reviews and scientific studies published in the literature explain that the LTI is essentially a transition region with steep gradients in altitude: the dominance of the neutral atmosphere decreases within the LTI as evidenced by the decrease of the neutral density and the drastic increase of the temperature due to absorption of solar EUV radiation that occurs within the thermosphere. On the other hand, this is also the region where near-Earth space, controlled by electromagnetic effects, starts to influence the overall dynamics as part of the

neutrals are dissociated and the medium has the characteristics of a plasma system. First and foremost, this suggests that the LTI is a region where the underlying physical processes change in nature, warranting understanding both from the atmospheric perspective as well as in terms of space plasma physics.

In the Earth's denser lower atmospheric regions, up to the mesopause around 90 km altitude, the motion of the atmosphere is driven by both solar irradiance and waves. The dynamics is typically described as a flow governed by the laws of continuum

fluid dynamics, for a gaseous fluid that is electrically neutral. In the continuum assumption, averaging is performed over sampling volumes, such that the fluid particles are normally distributed and can thus be described in terms of local bulk macroscopic properties, notably pressure, temperature, density and flow velocity. The continuum assumption requires the sampling volume to be in thermodynamic equilibrium, which implies a high frequency of collisions between atmospheric particles. Atmospheric flow is then predicted by solving the fundamental conservation equations including the conservation of

mass, momentum, and energy, and a thermodynamic equation of state. The energy from solar irradiance is mostly deposited as sensible and latent heat fluxes, and via direct absorption of shortwave (solar) radiative energy, for instance by ozone in the ozone layer and of re-radiated energy, typically by greenhouse gases and clouds.

Above the mesopause, neutral densities become so low that collisions gradually become less important, while the density of the electrically charged ionospheric plasma increases. In contrast to the atmospheric material, near-Earth space plasmas cannot

be represented by a similar continuum assumption due to the scarcity of collisions. The laws controlling plasma motion need to be incremented by electromagnetic forces, and thus the forcing from the magnetosphere needs to be taken into account. At high latitudes, the ionosphere is coupled via the magnetic field to the magnetosphere, and even further into the solar wind. Further, plasma particles are typically not normally distributed, implying that plasmas cannot be described by, e.g., a single temperature. In the transition region between the atmosphere described by the continuum dynamics, and geospace described

by plasma kinetic theory, at altitudes roughly between 80 and 200 km, the atmosphere starts to be significantly affected by the presence of the ionosphere. The neutral particles and plasmas interact through collisions and charge exchange, which maximise at altitudes between 100 and 200 km but remain important up to around 500 km altitude, the nominal base of the exosphere, beyond which collisions are practically non-existent.

Even though the LTI is characterised as a transition region between the atmosphere and space, it is also markedly a region

having characteristics of its own. This is particularly true in terms of the energy sink that the region represents. From the atmospheric perspective, the energy of upward propagating atmospheric waves, such as planetary waves, tides, and gravity waves (for a review, see Vincent, 2015) is deposited into the LTI. These waves can drive plasma instabilities, which in turn lead





to small-scale variations that can cause disruption of radio signals (e.g., Xiong et al., 2016). On the other hand, at polar latitudes, the LTI is a major sink of energy transferred from the solar wind by processes within the magnetosphere and ionosphere, which

are not well understood. In particular, during times of very large solar and geomagnetic activity, for example as a response to interplanetary coronal mass ejections (ICMEs, e.g., Richardson and Cane, 2010) and stream interaction regions (SIRs) followed by high-speed streams (HSSs, e.g., Grandin et al., 2019a), this energy input increases substantially, and can represent a larger energy source than that provided by solar irradiance. Thus, the energetics, dynamics, and chemistry of the LTI result from a complex interplay of processes with coupling both to the magnetosphere above and to the atmosphere below.

The neutral–plasma interactions and dynamics within the LTI are poorly understood, mainly due to a lack of systematic observations of the key parameters in the region. In the case of scarce observations, the solution is usually to build a model which can be used to obtain information on the region. However, in the case of the LTI, this approach has been markedly difficult due to the complexity of the system: the atmospheric models solving general circulation, chemistry, or the climate system (e.g., Gettelman et al., 2019) normally do not take into account electromagnetic forces. On the other hand, the magnetosphere

models using a first-principles plasma approach, either using the magnetohydrodynamics description (MHD; e.g., Janhunen et al., 2012; Glocer et al., 2013) or the (hybrid-)kinetic description (e.g., Omidi et al., 2011; Palmroth et al., 2018), have to be coupled to the ionosphere and neutral atmosphere. The ionospheric first-principles (e.g., Marchaudon and Blelly, 2015; Verronen et al., 2005) or (semi-)empirical models (e.g., Bilitza and Reinisch, 2008) require coupling both to the magnetosphere and to the atmosphere. In the recent years, the different modelling communities have started to integrate the dedicated models

towards new regimes, as, e.g., the Whole Atmosphere Community Climate Model (WACCM) has been extended to cover the thermosphere and ionosphere to about 500 km altitude (WACCM-X Liu et al., 2018a). The MHD-based magnetospheric models have been coupled to the ionosphere and neutral atmosphere (Tóth et al., 2005). However, even though the models may currently be the main tool used to provide information on the coupled system, they can only be trusted after careful validation and verification. Hence, ultimately the only way to understand the LTI holistically is by acquiring systematic measurements of

the system.

There is a growing recognition that the Earth needs to be studied and understood as a coupled system of its various components. The European Space Agency's (ESA) Living Planet Programme embraces this need, calling for studies of the many linkages within the system. From this viewpoint, it follows that our understanding is only ever as good as the weakest link. One such weak link currently is the connection between the Earth and space. For example, there are considerable changes caused

by currents and energetic particles from outer space impinging on the atmosphere, and some of these changes are not well sampled and quantified at all, leading to significant (and maybe even critical) uncertainties. ESA's Earth Explorer 10 candidate mission Daedalus (Sarris et al., 2019) has been designed to explore the LTI systematically for the first time *in situ* to address the challenges within the LTI described above.

This paper introduces the science behind the Daedalus candidate mission. First, we list the three main outstanding topics

under research, related to the LTI energy balance, LTI variability and dynamics, and LTI chemistry. The logic of the paper is to present the outstanding science questions first with a short summarising background. These science topics lead to the need of observing the key LTI parameters, which are divided into those that can be observed directly, and those that need to be derived





from several other parameters. The bulk of the review concentrates into these direct and derived observables, while the science questions are on purpose concise, giving only a few central literature references. An important choice made in this paper is

related to the most important energy deposition mechanism driven by the solar wind and magnetospheric forcing, called *Joule heating*. This is such a vast topic that it requires a review of its own. However, here the emphasis is on the parameters required to assess Joule heating accurately.

In the two most recent review papers of the LTI, Sarris (2019) emphasises the main gaps in the current understanding of this key atmospheric region and discusses the related roadmaps and statements made by several agencies and other international

bodies, whereas Heelis and Maute (2020) provide a detailed review of the physical processes and couplings within the LTI. The purpose of this paper, in turn, is to systematically list and discuss the parameters that can be observed or derived from *in situ* measurements, underlining the state-of-the-art in observations and numerical models. The intention is to give a background for the measurement setup of any given future mission within the LTI, from the viewpoint of the major outstanding questions. The paper is organised as follows: Section 2 presents the outstanding science questions related to the LTI. Sections 3 and 4 review

the current understanding of the LTI observed and derived parameters, respectively, which are key to improve the understanding of the region and required to close the outstanding questions. Section 5 ends the paper with concluding remarks.

## 2   Open questions in LTI energetics, dynamics, and chemistry

To assess the role of the LTI as a crucial component within the Earth's atmospheric system, it is important to understand the dominant processes involved in determining the energetics, dynamics, and chemistry of the LTI. Such knowledge is also critical

to develop capabilities to specify and forecast space weather phenomena that occur, originate and are modified in this region. This section summarises these broad topics to give the background for the required observed and derived parameters outlined later. Figures 1 and 2 illustrate some of the crucial parameters in terms of energetics (temperatures on Fig. 1a and 2a), dynamics (neutral winds and ion drifts on Fig. 1b–c and 2b), and chemistry (electron density on Fig. 1d and 2c, and neutral density on Fig. 2c), both over global scales in Fig. 1 and as altitude profiles at selected latitudes on the duskside in Fig. 2. In particular,

Fig. 2 reminds the usual nomenclature used in atmospheric and ionospheric studies, indicating the names of the atmospheric layers alongside the temperature profiles (Fig. 2a) and the *D*, *E*, *F* regions (or layers) in the ionosphere alongside the electron density profiles (Fig. 2c).

### 2.1   LTI energetics

In the following, LTI energetics refers to the energy input, deposition, dissipation, and in general the energy balance within

the LTI. Energetics is driven on the one hand by the solar radiative flux and on the other hand by energy deposition into the LTI from above (near-Earth space) and below (lower atmospheric regions). The solar radiative flux is mostly controlled by the inclination of the planet's rotation axis with respect to the Sun-Earth line, as well as by the distance from the Sun. The energy input from below mainly consists of atmospheric waves propagating upwards. The energy input from above is extracted from the solar wind and processed by the magnetosphere, and it affects, e.g., the motion of the ionospheric charged particles



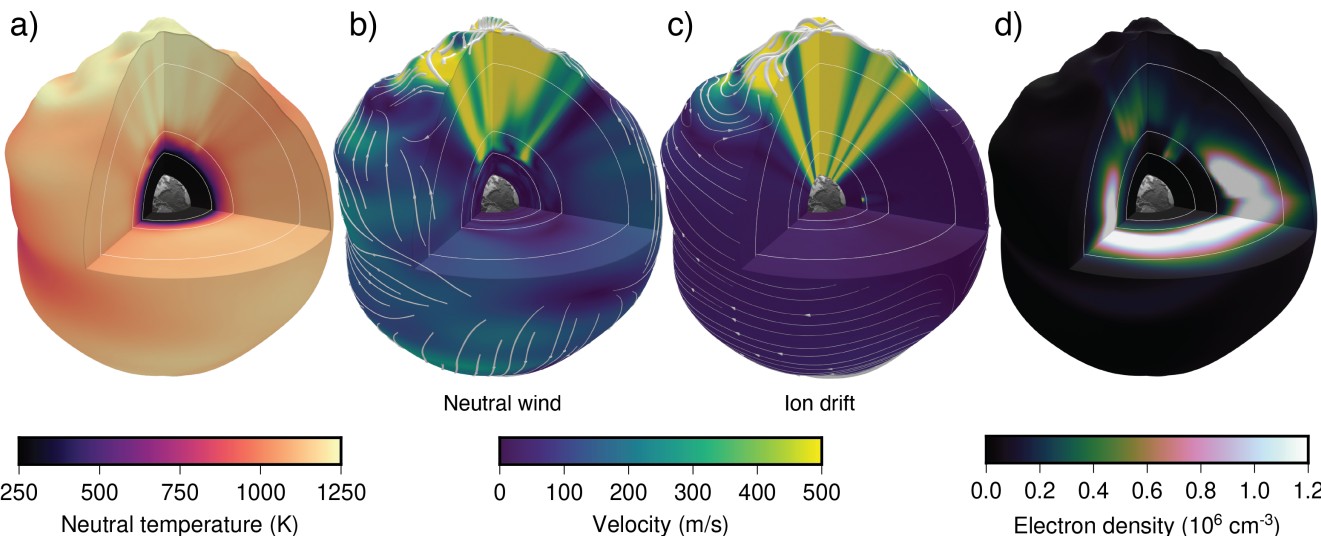

**Figure 1.** Overview of some key atmosphere/ionosphere parameters from the WACCM-X model: a) neutral temperature, b) neutral wind magnitude and streamlines, c) ion drift magnitude and streamlines and d) electron density. The model output is from a simulation of the 2015 St. Patrick's Day storm, showing the simulated state of the atmosphere on 17 March 2015, 18:00 UTC, during a period of significant high-latitude energy input. Slices through the model are shown at the top pressure level, at 0° and −90° longitude and at −1° latitude. The meridional slice over the Greenwich meridian (top right of each sphere) shows a dusk profile, while the 90° west slice shows a noon profile. Pressure level geopotential heights from the model have been exaggerated by 50 times to show vertical detail.

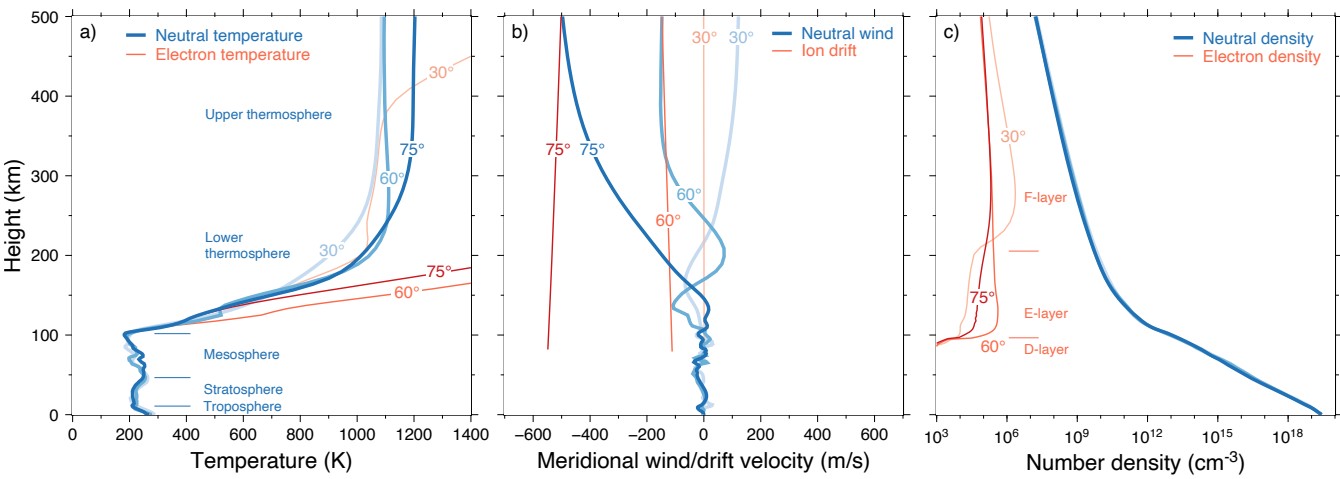

**Figure 2.** Altitude profiles from the WACCM-X model of some key atmospheric parameters at three different geographic latitudes (30, 60 and 75° N), along the Greenwich meridian at 18:00 UTC on 17 March 2015, consistent with the vertical/meridional slice through this model at that location in Figure 1.





and electromagnetic fields within the LTI (e.g., Palmroth et al., 2004). There are two primary energy sinks which deposit magnetospheric energy into the ionosphere: Joule heating (JH), and particle (electron and proton) precipitation, of which the current understanding suggests that JH represents the larger sink (e.g., Knipp et al., 1998; Lu et al., 1998). However, currently the energy deposited per unit volume at LTI altitudes via JH and particle precipitation is not known. Furthermore, the influence of this energy deposition on the local transport, thermal structure, and composition within LTI altitudes is also poorly known.

### 2.1.1 Joule heating

Joule heating is in general terms caused by electric currents flowing through a resistive medium, which causes heating within the medium. In the geospace, the current system consists of field-aligned currents (FACs, see Sect. 4.2), which find their closure through ionospheric horizontal currents in the ionosphere (e.g., Sergeev et al., 1996), which is a resistive medium as neutral and charged particles undergo collisions. Ultimately, the power density dissipated by JH is according to Poynting's theorem

$\mathbf{j} \cdot \mathbf{E}$ where $\mathbf{j}$ is the electric current density, and $\mathbf{E}$ the electric field in the frame of the neutral gas. The electric field in other reference frames with the neutral gas velocity $\mathbf{U} \neq \mathbf{0}$ is $\mathbf{E}' = \mathbf{E} + \mathbf{U} \times \mathbf{B}$ with $\mathbf{B}$ the magnetic field (Kelley, 2009).

In this primary energy deposition mechanism, the additional energy from the magnetosphere forces the plasma to advect relative to the neutral gas, leading to ion–neutral frictional (or Joule) heating. During geomagnetic storms, current knowledge indicates, albeit with large uncertainties, that this energy sink is on a par with the heat created by absorption of solar radiation,

which otherwise is the major driver of atmospheric dynamics (Knipp et al., 2005). The effects of moderate to strong geomagnetic activity can be significant in mid and equatorial latitudes, as auroral JH can launch travelling ionospheric disturbances which can have measurable effects down to equatorial latitudes (e.g., Zhou et al., 2016; de Jesus et al., 2016). By enhancing the ion temperature, JH modifies the chemical reaction rates and thus the local chemical equilibrium and ion and neutral composition. The way by which neutral winds, ion drifts and electric fields interplay to generate heating is largely unknown, primarily

due to the lack of co-located measurements of all key parameters involved. Since the topic of JH is vast, it is left as a subject of a subsequent paper, while we cover some of the knowledge and open questions of JH in Sect. 4.7.

To understand this energy deposition mechanism, it is imperative to explore the energy deposited into the LTI through JH, by simultaneously measuring the comprehensive set of variables determining JH in the auroral latitudes and 100–200 km altitude regions where it maximises, sampled over a broad range of atmospheric and geomagnetic conditions, and at a resolution that

captures the key scales associated with this heating process. In particular, since the quantification of JH is significantly affected by the measurements of the Pedersen conductivity (e.g., Palmroth et al., 2005), it is necessary to characterise how collision cross-sections, frequencies, and the resulting conductivities vary with altitude and conditions in the LTI, by simultaneously measuring the comprehensive set of variables determining these parameters over the relevant altitudes and for a range of atmospheric and geomagnetic conditions.

### 2.1.2 Precipitation-driven energy input

The second most important energy deposition mechanism is caused by particle precipitation, which is typically divided into two categories: lower-energy auroral ($\sim$0.01–20 keV, mostly electron) precipitation, depositing energy within $\sim$100–300 km



altitude, and energetic (>30 keV) particle precipitation (EPP), including relativistic (>1 MeV) energies, consisting of energetic electrons and ions depositing energy below ∼90 km altitude (Berger et al., 1970). Auroral ion precipitation occurs, with

the specificity of its own that precipitating protons can undergo multiple charge-exchange interactions with atmospheric constituents on their way down, leading to a spreading of the affected area (see special section by Galand, 2001). The sources for auroral precipitation and EPP are particles both directly coming from the Sun or accelerated by various processes in the magnetosphere. Broadly speaking, auroral precipitation comprises larger number fluxes (Newell et al., 2009), while EPP consists of higher energies. Hence both affect the energy deposition within the LTI, the former through larger areas and the latter

through higher energies.

The energy input from particle precipitation is given by the energy of the incoming particles deposited via either dynamical or chemical processes at the altitude of dissipation, for example through electron temperature enhancement, ionisation of neutrals, excitation of neutrals or ions, and dissociation of molecular species producing chemical components (see also Sect. 2.3). The altitude of maximum energy deposition by precipitation is determined by particle energies (e.g., Turunen et al.,

2009, Figure 3): Relativistic ions ($E > 30$ MeV) and electrons ($E > 1$ MeV) penetrate down to the stratosphere, energetic ions ($1 < E < 30$ MeV) and electrons ($30 < E < 1000$ keV) deposit their energy through ionisation into the mesosphere, while the lower-energy ions ($E < 1$ MeV) and auroral electrons ($E < 30$ keV) impact the thermosphere. The local values of precipitation-induced heating are however largely unknown, as its quantification proves challenging due to the scarcity of suitable observations. Detailed measurements of the energy spectrum and flux of particles passing through the thermosphere as

a function of solar/geomagnetic conditions are key to accurately quantifying the impact of precipitation on the climate system (see Sect. 2.3.1).

To quantify the energy deposited into the LTI through particle precipitation, it is necessary to measure the energy spectrum and flux of precipitating particles at the auroral latitudes in regions where it maximises, sampled over a broad range of geomagnetic conditions, and at resolutions in energy and pitch-angle that capture the characteristic scales associated with the

heating, ionisation, and dissociation processes of interest. To assess the local response and relative importance of JH and EPP in the LTI, it is necessary to simultaneously measure the comprehensive set of corresponding changes in composition, flows, and temperatures with adequate temporal resolution to capture the involved processes.

### 2.1.3 Energetics driven by the neutral atmosphere

There are several types of waves within the lower atmosphere which travel vertically towards the LTI and are expected to

dissipate there. These waves couple with the neutral wind, temperature field, and density in the LTI. They are also believed to seed plasma instabilities, especially in the low-latitude region, and the upward propagating waves can produce large shears that may affect the overall circulation within the LTI. In particular, gravity waves (see Sect. 4.9) contribute significantly to the LTI energetics. From a high-resolution Whole Atmosphere Community Climate Model (WACCM) simulation (with horizontal resolution of ∼25 km, Liu et al., 2014), it has been calculated that the total upward energy flux by resolved waves at 100 km

is 100–150 GW (Liu, 2016), which is comparable to the daily average JH power input (Knipp et al., 2004). This is likely an underestimation of the actual energy flux by gravity waves, since waves with horizontal scales less than 200 km are poorly





resolved due to numerical dissipation (or not resolved at all) in the model. The energy deposition rate is also estimated based on parametrised gravity waves: total wave energy deposition rate at 100 km is 35 GW, and 75% of that comes from parametrised and resolved gravity waves (Becker, 2017). The role of the neutral atmosphere forcing for LTI energetics can only be estimated, because there are no comprehensive and systematic observations of the coupling between neutral and ions in the LTI.

## 2.2 LTI variability and dynamics

This section summarises typical phenomena of spatial and temporal variability in the LTI region and mentions dynamical processes that lead to reorganisation of, e.g., neutral or electron density, conductivity, or wind. We consider forcing from above, defined as variations driven by magnetospheric dynamics (Sect. 2.2.1). In this topic, the current key scientific questions related to LTI variability and dynamics are to understand the ways in which the magnetosphere drives plasma motion in the high-latitude LTI, and how this motion affects the motion of the neutrals. We also consider forcing from below through atmospheric waves (Sect. 2.2.2). In this topic, the current key research question is to understand how large shears, sharp gradients, and small-scale plasma instabilities develop in the LTI in response to driving from below. LTI variability and dynamics take a special form at the low geomagnetic latitudes, summarised in Sect. 2.2.3. In low latitudes, the current key scientific question is to quantify the relative contributions of magnetospheric, solar and atmospheric forcing influencing LTI fluid dynamics and electrodynamics.

### 2.2.1 LTI forcing from above

Magnetospheric driving of the LTI can take the form of electromagnetic driving due to rapid variations in the geomagnetic field, and wave-particle interactions within the magnetosphere. Both processes involve the geomagnetic field (Sect. 3.6) and FACs (Sect. 4.2). The geomagnetic field variations are chiefly due to *substorms* which are often defined as periods of solar wind energy loading and subsequent magnetospheric unloading (e.g., McPherron, 1979). While there is still much debate about the sequence of events that lead to a substorm onset (e.g., Angelopoulos et al., 2008; Lui, 2009), from the phenomenological perspective it is agreed that substorms involve magnetotail reconnection (e.g., Angelopoulos et al., 2008), a FAC system connecting the tail plasma sheet to the ionosphere called substorm current wedge (e.g., Keiling et al., 2009), fast tail plasma flows (e.g., Angelopoulos et al., 1994), dipolarisation of the tail magnetic field (e.g., Runov et al., 2011), plasmoids launched tailwards (e.g., Ieda et al., 2001), and rapidly northward propagating bright auroral emissions (e.g., Frey et al., 2004). It is not within the scope of this paper to review all the substorm-related subtleties; rather our purpose here is to emphasise the role of substorms as one of the chief magnetospheric drivers of LTI energetics and dynamics. This driving is mostly manifested as increased precipitating particle fluxes as well as intensified FACs.

Another broad category of magnetospheric drivers of the LTI consists of the various waves which modify the proton and electron pitch angles such that the particles precipitate into the LTI. These waves have a multitude of drivers, and their characteristics and role in driving the LTI vary greatly. For example, Alfén waves, driven by solar wind-magnetosphere interactions, propagate into the LTI, transferring energy and momentum, as well as modifying LTI local plasma properties such as the density, temperature and conductance via the total electron content (e.g., Pilipenko et al., 2014; Belakhovsky et al., 2016). Various





wave modes, primarily ultra-low frequency (ULF), very-low frequency (VLF) and electromagnetic ion cyclotron (EMIC) waves
drive energetic particle precipitation (Thorne, 2010, see also Sect. 3.1), which drives chemistry changes in the LTI (Sect. 2.3).
The characteristics and propagation of these waves is an important unsolved problem; however, they are well-measured only on
the ground (e.g., Sciffer and Waters, 2002; Engebretson et al., 2018; Graf et al., 2013; Manninen et al., 2020) or above the LTI
around 400 km altitude (e.g., Li and Hudson, 2019). To build a complete picture of wave propagation through the LTI, direct
*in situ* measurements of these waves are required simultaneously with the plasma and neutral gas parameters, which determine
the wave propagation in this region.

Downward LTI forcing is not limited to processes originating from the magnetosphere. Solar flares are also known to
enhance electron density and hence JH in the LTI (e.g., Pudovkin and Sergeev, 1977; Sergeev, 1977; Curto et al., 1994;
Yamazaki and Maute, 2017). While the solar-flare-driven ionospheric current, or crochet current, near the subsolar region has
been intensively studied (e.g., Annadurai et al., 2018), its couterpart at high latitudes has been poorly understood for 40 years,
although it significantly enhances pre-existing JH at the auroral electrojets (Pudovkin and Sergeev, 1977). The modification of
the auroral electrojets by solar flares can be more than a mere enhancement, as it can also lead to a change in the direction of
the electrojet, and the resulting geomagnetic deviation can exceed 200 nT. It is quite possible that the altitude of the ionospheric
current also changes, but no measurement method to prove this has been proposed.

To understand the forcing from above, it is necessary to explore the momentum transfer between the plasma and the neutral
fluid in the LTI, by simultaneously measuring the comprehensive set of variables determining the forces globally, sampling a
broad range of atmospheric and geomagnetic conditions, and over timescales that capture the involved processes.

### 2.2.2    LTI forcing from below

Ionised gas under the influence of the geomagnetic field affects greatly the overall dynamics of the LTI, which makes it distinct,
but not decoupled, from the atmosphere. In addition to Joule heating, the electromagnetic coupling asserts the Lorentz force
acting on the ionised gas, providing geospace with a lever on the atmosphere, and also providing a lever between hemispheres
connected by the dipolar geomagnetic field. Furthermore, the electromagnetic forcing affects and is affected by atmospheric
variations and disturbances, e.g., by planetary (Rossby) waves, gravity waves, and solar or lunar tides, originating from below
the LTI and propagating upwards. Many outstanding issues remain in our understanding of the complex large-scale and global
interactions between these processes and forces that act together to determine LTI dynamics. Especially the occurrence of
strong flow shears, steep gradients or rapid variations in the LTI parameters have been observed but not been studied systemat-
ically due to a lack of consistent measurements of the relevant parameters. Consequently, the effects of such structures on the
LTI dynamics are not well known. The physics of the different atmospheric waves is reviewed in Sect. 4.9.

To understand the driving from below, it is necessary to simultaneously measure all the variables defining not only the neutral
dynamics but also the electrodynamics as well as the corresponding local changes in composition, densities, and temperatures
at the relevant latitudes and altitudes, sampled over a range of atmospheric and geomagnetic conditions, and at temporal
scales that capture the key processes involved, including gravity waves, planetary waves and tides originating from the lower
atmosphere.





### 2.2.3 Variability and dynamics in the low-latitude LTI

At low latitudes, the dynamics of the LTI, comprising the ionosphere *E* region and lower *F* region, determines significant parts of the variability of the entire thermosphere and ionosphere through global electric field variations (Scherliess and Fejer, 1999) and related large- to medium-scale plasma transport, the most important phenomenon being known as the Equatorial Ionisation Anomaly (e.g., Walker et al., 1994; Stolle et al., 2008b). The *E* region dynamo which results from charged particles transported by thermospheric winds through the nearly horizontal magnetic field (e.g., Heelis, 2004) is understood to play a key role in

driving the electric fields and the equatorial electrojet, the latter being a ribbon of strong eastward dayside current flowing along the magnetic equator. While the general principles are described, the significant day-to-day variability of their magnitudes is still subject of investigation (e.g., Yamazaki and Maute, 2017). A special category of the LTI variability and dynamics within the low latitudes are post-sunset *F*-region equatorial plasma irregularities, in which the LTI and lower *F* region are believed to play an important role. Suggested initial perturbations for these plasma irregularities are the variability of the vertical plasma

drift at sunset hours (e.g., Huang, 2018; Wu, 2015; Stolle et al., 2008a) and the role of upward propagating gravity waves (e.g., Krall et al., 2013; Hysell et al., 2014; Yokoyama et al., 2019). The resulting *F*-region plasma irregularities cause severe effects on trans-ionospheric radio wave signal propagation (e.g., Xiong et al., 2016), and are thus an important source for space weather disturbances.

To understand the LTI behaviour within low latitudes, it is imperative to reveal the morphology of flow shears and sharp

gradients in the LTI and their role in driving plasma instabilities by simultaneously measuring the comprehensive set of variables that fully describe the LTI, including the ionospheric plasma density at a horizontal resolution that captures the relevant processes, sampled over a wide range of latitudes and altitudes, with sufficient time resolution, and over appropriate temporal scales for these processes.

## 2.3 LTI chemistry

The chemical composition of the LTI may change in response to particle precipitation (Sect. 2.3.1), temperature increase associated with frictional/Joule heating (Sect. 2.3.2), and through chemical heating (Sect. 2.3.3) resulting from exothermic reactions. The current key science questions in upper atmospheric chemistry are related to the chemical effects of EPP within the mesosphere (and the stratosphere below) as a function of geomagnetic driving conditions. Further, the role of driving conditions from below, including the upward-propagating gravity waves, in influencing the LTI chemistry is not known. Finally,

it is not known whether the current model boundary conditions (see below) provide a good representation of the LTI physics as a function of LTI conditions and solar activity. This section is dedicated to summarise the background to these topics.

### 2.3.1 Precipitation-driven chemistry

Electron and ion precipitation ionise and dissociate neutrals through collisions (Sinnhuber et al., 2012). This has a direct effect in the atmospheric chemical composition via ion chemistry which leads to production of odd hydrogen ($HO_x$) and

nitrogen ($NO_x$) (e.g., Codrescu et al., 1997; Seppälä et al., 2015). Considering the LTI coupling to lower atmosphere, odd





nitrogen ($NO_x = NO + NO_2$) is particularly important, because it has a long ($\sim$months) chemical lifetime in polar winter conditions, and it descends to mesospheric and stratospheric altitudes down to $\sim$35 km (Randall, 2007; Funke et al., 2014; Päivärinta et al., 2016) and catalytically destroys ozone (Damiani et al., 2016; Andersson et al., 2018). Ozone is an effective absorber of solar ultraviolet radiation, and its variability modulates the thermal balance of the middle atmosphere and polar

vortex dynamics (Brasseur and Solomon, 2005). These perturbations can propagate to surface levels and modulate regional patterns of temperatures and pressures (Gray et al., 2010; Seppälä et al., 2014). Investigation of atmospheric reanalysis datasets and coupled-climate model runs have shown that $NO_x$ and $HO_x$ have the potential to modify regional winter-time surface temperatures by as much as $\pm 5$ K, through re-distributing annular mode patterns at mid to high latitudes in both northern and southern hemispheres (Seppälä et al., 2009; Baumgaertner et al., 2011). To understand these questions, it is necessary to make

simultaneous observations of the EPP flux, energy spectral gradients, ion composition, and $NO_x$, and measure EPP fluxes with good resolution in the energy and pitch angle. Further, the involved energy spectral gradients need to be described, along with the energy ranges that cover the deposition altitudes from the lower thermosphere to the mesosphere down to the stratopause. These measurements need to be sampled at rates fast enough to resolve different precipitation mechanisms and boundaries on scales of 10 km or smaller.

The LTI region chemistry is recognised to be important for long-term climate simulations due to its role in solar-driven $NO_x$ production and ozone impact (Matthes et al., 2017). However, there are substantial differences between simulated and observed distributions of polar $NO_x$, owing partly to an incomplete representation of electron precipitation (Randall et al., 2015). Further, adequate climate simulations require a $NO_x$ upper boundary condition as well as a representation for the dynamical-chemical coupling between thermospheric $NO_x$ and stratospheric ozone. For so-called high-top models, with upper

boundary in the thermosphere, the boundary conditions can be defined by empirical models based on satellite data (e.g., Marsh et al., 2004), which depend on geomagnetic indices, day of the year, and solar flux. However, current models are based on temporally limited data and do not cover full solar cycles and/or differences between solar cycles, and recent studies indicate a need for improvements (Hendrickx et al., 2018; Kiviranta et al., 2018). To improve the model boundary conditions, it is necessary to make observations of $NO_x$ in the polar lower mesosphere below 150 km to characterise the NO reservoir and

variability. Preferably, the measurements should be carried out long enough to cover the solar cycle, and different EPP events to improve understanding of the drivers for the climate model boundary conditions.

### 2.3.2 Heating-driven chemistry

Changes in the ion and neutral temperatures, for instance associated with ion–neutral frictional heating, affect the chemical reaction rates in the LTI and can consequently modify the LTI composition. Grandin et al. (2015) found that during high-

speed-stream-driven geomagnetic storms the auroral-oval $F$-region peak electron density can decrease by up to 40% in the evening magnetic local time (MLT) sector, especially around the equinoxes. The suggested mechanism to account for this electron density decrease is that ion–neutral frictional heating associated with substorm activity may increase the ion and neutral temperatures on timescales much less than an hour, resulting in an enhancement of the electron loss rate by increasing both the chemical reaction rates (functions of the ion temperature) and the molecular densities by upwelling of the neutral




atmosphere associated with the neutral temperature increase. A subsequent study by Marchaudon et al. (2018) confirmed that this mechanism, especially through the latter process, can account for the long-lasting $F$-region peak reduction. Heating-driven composition changes in the LTI have also been revealed in association with subauroral polarisation streams (SAPS; e.g., Wang et al., 2012) and solar proton events (e.g., Roble et al., 1987). However, not many studies discuss heating-driven chemistry in the LTI, indicating a lack of systematic measurements.

### 2.3.3 LTI Chemistry and Chemical Heating

Chemical heating is one of the main energy sources in the LTI, together with Joule heating, EUV radiation and particle precipitation heating, resulting from the storage in latent chemical form and subsequent release of energy (Beig, 2003; Beig et al., 2008). Chemical heating influences the upper atmosphere in a variety of ways, including the formation of mesospheric inversion layers (Ramesh et al., 2013). Chemical energy is deposited in the LTI through the exothermic reactions typically

involving oxygen (atomic and molecular) and ozone (e.g., Singh and Pallamraju, 2018). Neutral species, namely $O_3$, $H_2O$, $CO_2$, OH, and aerosols are believed to play a role both in the chemistry of the LTI and in the radiative balance of the mesosphere (Mlynczak, 2000). On the other hand, $CO_2$ molecules can induce radiative cooling in the LTI through their emission at 15 μm. Especially between 75 and 110 km altitude, this emission is the only significant cooling mechanism (e.g., Fomichev et al., 1986), while below, radiative cooling by ozone and $H_2O$ is also important (e.g., Bi et al., 2011). Quantifying the contribution

of chemical heating to the changes in the LTI composition is vital in order to understand the full radiative balance of the upper atmosphere. Furthermore, the spatial and temporal distributions of neutral species could be used as tracers of wave and tidal phenomena (Solomon and Roble, 2015), which affect the overall dynamics of the LTI. Therefore it is important to obtain measurements of the chemical composition and heating in the LTI.

## 3 Observed LTI parameters: Current understanding

### 3.1 Precipitating particle fluxes and energies

Particle precipitation is very much connected to the overall electrodynamic coupling within the LTI. Precipitation leads to increased ionospheric conductivities (Aksnes et al., 2004) and creates field-aligned currents (FACs, see Sect. 4.2). FACs close in the $E$ region of the ionosphere, leading to ion-neutral frictional heating (Millward et al., 1999; Redmon et al., 2017, see Sect. 4.7). Since it plays such a leading role in the electrodynamic coupling, we discuss precipitation first.

Particles (electrons and ions) precipitate into the LTI when they are scattered into the bounce loss cone. Pitch-angle scattering can be due to the magnetic field curvature radius being close to the particle gyroradius (Sergeev and Tsyganenko, 1982) or to wave–particle interactions. For instance, lower-band chorus waves, often present in the morningside and dayside magnetosphere, can lead to energetic ($E > 30$ keV) electron precipitation (Thorne et al., 2010), whereas electromagnetic ion cyclotron (EMIC) waves can be efficient in scattering kiloelectronvolt protons and megaelectronvolt electrons into the bounce loss cone

(Rodger et al., 2008; Yahnin et al., 2009). Other suggested pitch-angle scattering waves include the plasmaspheric hiss, which



may contribute to the precipitation of subrelativistic electrons (He et al., 2018). Phenomena such as pulsating aurora have been found to be associated with precipitating electrons across a wide range of energies (e.g., Grandin et al., 2017b; Tsuchiya et al., 2018), which suggests interaction with whistler chorus waves (Miyoshi et al., 2015) or electrostatic electron cyclotron harmonic waves (Fukizawa et al., 2018). Evaluating the relative contribution of each scattering process to the global precipitation

budget is challenging; obtaining particle measurements at multiple pitch angle values in the bounce loss cone with good energy resolution across the energy range could prove decisive in this endeavour.

Precipitating particles can have energies ranging from tens of electronvolt to tens of megaelectronvolt. While low-energy ($E \approx 0.1 - 30$ keV) electrons and protons primarily precipitate at high latitudes, in the polar cusps and in the nightside auroral oval (which is usually above $\sim 65°$ geomagnetic latitude), relativistic electrons from the outer radiation belt ($E \approx$

$0.1 - 10$ MeV) mostly precipitate at subauroral latitudes, i.e., equatorwards from the auroral oval. Solar energetic particles ($E > 10$ MeV protons), on the other hand, precipitate directly from the solar wind into the polar region (geomagnetic latitudes above $\sim 60°$); however, the largest of those events are rare and typically occur only a few times per solar cycle (Neale et al., 2013). Energetic neutral atoms (1–1000 keV, principally within the 100 keV range; Orsini et al., 1994; Roelof, 1997; Goldstein and McComas, 2013) are produced via charge exchange when energetic ions interact with background neutral atoms such as

Earth's geocorona. They can play a role in mass and energy transfer to lower latitudes beyond the auroral zone (Fok et al., 2003) and become strongly coupled to precipitating energetic ions in the lower thermosphere (Roelof, 1997).

To date the most comprehensive measurements of particle distributions in the near-Earth environment have been made by flagship spacecraft missions such as DEMETER (Sauvaud et al., 2006), Cluster (Escoubet et al., 2001), Magnetospheric Multiscale (MMS; Burch et al., 2016), Arase (Miyoshi et al., 2018), and the Van Allen Probes (Mauk et al., 2013). However,

at high altitudes, bounce loss cone angles have values of the order of a few degrees only, which is too small to be resolved by most particle instruments carried by those spacecraft. On the other hand, at altitudes where low-Earth orbit (LEO) satellites fly, the bounce loss cone at auroral latitudes has its edges at an angle of about $60°$ from the magnetic field direction (Rodger et al., 2010a); it is therefore possible to resolve it with particle detectors. A large number of LEO spacecraft missions have flown particle detectors measuring differential and integral precipitation fluxes. The Solar, Anomalous, and Magnetospheric

Particle Explorer (SAMPEX; Baker et al., 1993) mission (1992–2012) produced megaelectronvolt electron precipitation data that have been used in scientific studies (e.g., Blum et al., 2015). The SSJ experiment aboard Defense Meteorological Satellite Program (DMSP) satellites has provided precipitating proton and electron observations in up to 20 channels covering the lower energies (30 eV–30 keV) since 1974 (e.g. Hardy et al., 1984; Redmon et al., 2017); Figure 3a gives an example of differential number flux of precipitating electrons measured by DMSP-F18 on 20 January 2016 in the evening sector of the

northern auroral oval. Higher-energy ($> 30$ keV) precipitation observations have on the other hand been routinely provided by NOAA Polar-orbiting Operational Environmental Satellite Space Environment Monitor (POES/SEM) instrument suite since 1979, although measurements have suffered from contamination issues that were corrected by Asikainen and Mursula (2013). Particle detectors can nowadays even be included in nanosatellite missions; one example of upcoming CubeSat missions aimed to measure particle precipitation is FORESAIL-1 (Palmroth et al., 2019) which is expected to measure energetic and relativistic

electrons and protons.



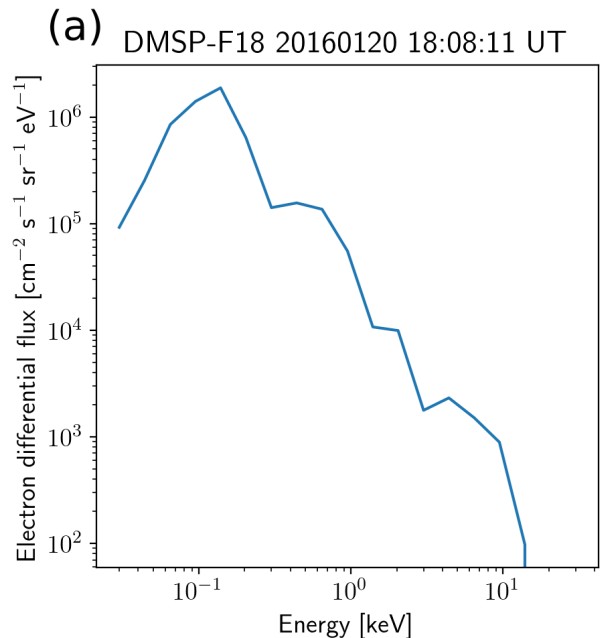

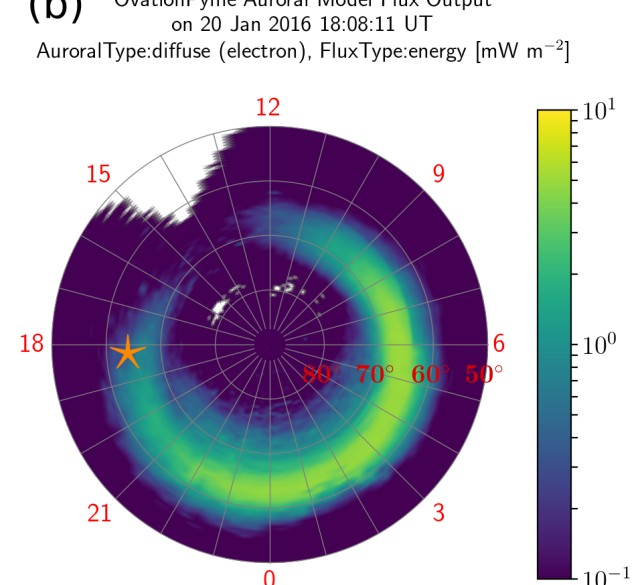

**Figure 3.** (a) Auroral electron precipitation differential number flux measured by the DMSP-F18 spacecraft on 20 January 2016 at 18:08:11 UT. (b) Map of diffuse auroral electron energy flux in the northern hemisphere given by the OVATION-Prime model at the same time. The radial coordinate is geomagnetic latitude, and the angular coordinate is MLT. The orange star indicates the position of the DMSP-F18 spacecraft.

Indirect observations of particle precipitation can be achieved through various types of observations. Balloon experiments flying in the stratosphere can detect Bremsstrahlung emission produced by precipitating particles interacting with neutrals in the atmosphere, as is done during BARREL campaigns (Woodger et al., 2015). Energetic electron precipitation is routinely monitored from the ground using riometers, which measure the cosmic noise absorption in the D region of the ionosphere
associated with particle precipitation (e.g., Hargreaves, 1969; Rodger et al., 2013; Grandin et al., 2017a). Phase and amplitude perturbations to subionospheric man-made narrow-band transmitter signals propagating over long distances are also routinely used to identify energetic electron precipitation (Clilverd et al., 2009). Incoherent scatter radar observations can be used to retrieve precipitating electron energy spectra (Virtanen et al., 2018), and to monitor the ionospheric impact of particle precipitation (Verronen et al., 2015).

Empirical models have been developed by deriving statistical patterns of particle precipitation as a function of geomagnetic activity based on several years of spacecraft observations. The Hardy model (Hardy et al., 1985, 1989) was established by compiling two years of DMSP measurements of precipitation, and provides differential number fluxes of precipitating electrons and protons as a function of the $K_p$ index. More recently, the OVATION-Prime model (Newell et al., 2014) was developed to predict auroral power as a function of solar wind parameters. This model separates auroral precipitation into four types (dif-





fuse, monoenergetic, broadband and ion); Figure 3b gives an example of output of the diffuse auroral precipitation, obtained

during the conditions when the differential flux shown in Figure 3a was observed. For higher energies, while the AE-8 (elec-

trons) and AP-8 (protons) maps provide trapped fluxes in the radiation belts (Vette, 1992), the models developed by van de

Kamp et al. (2016) and van de Kamp et al. (2018) predict 30–1000 keV electron precipitation fluxes as a function of the $A_p$

index based on analysing energetic electron precipitation observed by POES satellites during 1998–2012. Such climatologies

prove particularly useful for space weather predictions and can be used as inputs to ionospheric models, such as the IRAP

Plasmasphere-Ionosphere Model (IPIM; Marchaudon and Blelly, 2015) or the Whole Atmosphere Community Climate Model

(WACCM3; Kinnison et al., 2007). Finally, a few attempts to model particle precipitation in global, first-principle simulations

of the near-Earth environment have been made, in magnetohydrodynamics models (e.g., Palmroth et al., 2006a), in some cases

coupled with a test-particle code (e.g., Connor et al., 2015), as well as in hybrid-particle-in-cell simulations (e.g., Omidi and

Sibeck, 2007) and more recently using a hybrid-Vlasov model (Grandin et al., 2019b).

## 3.2    Temperatures

The LTI temperature is a key background parameter, not only because it is a state parameter for the thermosphere itself, but

it is also key in ultimately driving neutral winds and atmospheric expansion, as well as determining conditions for chemical

reactions. While ion and electron temperatures, $T_i$ and $T_e$, can exceed the neutral temperature $T_n$ by thousands of kelvin (see

Figure 2a showing neutral and electron temperature profiles at selected latitudes obtained from a WACCM-X simulation), the

largest thermal energy reservoir in the LTI is in the neutral gas simply because of the low degree of ionisation in the LTI (see

Figure 4). The largest heat production is by absorption of solar EUV and UV radiation which is ionising and dissociating

molecules. This process accounts for the well-known basic vertical structure of $T_n$ and the thermospheric chemical composi-

tion.

Reliable measurements of $T_n$ have been difficult and less abundant compared to those of the neutral density itself where

especially the analysis of drag on satellite orbits has boosted the available data in the recent decades. In diffusive equilibrium

(for each gas component) the profiles of density and $T_n$ are not independent. Early models of the thermosphere were based on

this assumption and an empirical formula, sometimes called the Bates profile:

$$T_n(z) = T_\infty - (T_\infty - T_{z_0}) \exp\left(-\frac{z - z_0}{H}\right) \tag{1}$$

with $T_\infty$ the exospheric temperature, $T_{z_0}$ the temperature at the base, $z_0$ the height of the base, and $H$ a scale height (Bates,

1959). Sources of $T_n$ measurements include mass spectrometers on sounding rockets, which naturally are relatively sparse, on

satellites, which do not cover well the lower parts of the LTI, and by optical methods like UV occultations observed in space

and ground-based Fabry-Perot interferometers.

   Incoherent scatter radars (ISRs) can reliably measure $T_i$ when the mean ion mass is known or assumed. ISR measurements

of $T_i$ are a core resource for the construction of empirical models, particularly the widely used NRLMSIS-00 (Picone et al.,

2002). Below about 160 km altitude, the molecular ions $O_2^+$, $NO^+$, and $N_2^+$ with very similar masses are dominant, and in

the topside ionosphere the main ion is $O^+$. In these altitude regions, the $T_i$ estimation by ISR is based on relatively reliable





knowledge of the mean ion mass. During geomagnetically quiet times, after sunset, before sunrise, and preferably at mid and low latitudes, ion-neutral frictional heating is not expected to be significant. During geomagnetic activity, ion-neutral frictional

(Joule) heating (see Sect. 4.7), particle precipitation (see Sect. 3.1), and magnetic forcing ($\mathbf{j} \times \mathbf{B}$; see Sect. 4.1) increase, leading to atmospheric expansion and satellite drag (e.g., Liu and Lühr, 2005) and a general upwelling of the thermosphere. While diffusive equilibrium certainly cannot be assumed for a quantitative analysis in such dynamic situations, the upwelling must still be supported by substantial increases of $T_n$, as simulations have confirmed (Lei et al., 2010).

The thermospheric temperature can be increased significantly during large geomagnetic storms. In numerical simulations of

a major storm (November 8-10, 2004), $T_n$ was shown to increase from 750 K to up to about 1200 K at high latitudes, whereas at the equator the increase of $T_n$ over the quiet-time value, $\approx 1000$ K at 400 km height, never exceeded 30%, and was about 15% on average over the duration of the storm (Lei et al., 2010). The results and observations imply that the energy input into the thermosphere during this geomagnetic storm was invested for one part into geopotential energy, for another part into strong winds reaching a good fraction of the thermal velocity, and for a third part directly into heating of the neutral gas. Both the

potential and kinetic wind energy are eventually converted into heat, the latter by molecular viscosity which is important for the dynamics of the thermosphere. The relative contribution of each of these energy sinks during a strong geomagnetic storm requires further investigations to be determined in a quantitative way.

Compared to the solar-cycle-induced variation of $T_n$, the storm-induced changes seem to be still somewhat smaller. Typically $T_\infty$ varies between 750 and 1350 K over a solar cycle, with the power by solar EUV getting converted into both geopotential

energy of the atmosphere and directly into heat. The transition between solar-EUV-heated and dark regions is relatively smooth compared to the horizontal temperature gradients that are created by strong, localised Joule and particle precipitation heating. Therefore the latter probably also generate substantial "available potential energy" in the sense of Lorenz (1955).

### 3.3 Neutral and ion composition and densities

#### 3.3.1 Neutral and ion composition

The LTI is the region where the neutral atmosphere and the ionosphere are strongly coupled, and the exchange between neutrals and ions is continuous. This exchange occurs through ionisation and recombination, and is modulated by the solar UV flux, particle precipitation and the electrojets. The neutral and ion constituents have however very different scale heights and responses to the drivers as electrodynamic energy input, electric field, solar UV, atmospheric forcing, etc. (Schunk and Nagy, 1980). Figure 4 provides an example of density as a function of altitude for each of the major neutral and ion species in the

terrestrial upper atmosphere, for a given position and time. The ion densities are from the International Reference Ionosphere (IRI) model (Bilitza et al., 2014) and the neutral densities from the NRLMSISE-00 atmosphere model (Picone et al., 2002).

Composition observations are based on measuring the density of each species, ion or neutral, separately. The *in situ* composition measurements are performed by ion and neutral mass spectrometers, most notably onboard the AE-B and AE-C spacecraft (1966–1985, PI: H. C. Brinton) and onboard Dynamics Explorer-2 (1981–1983; Carignan et al., 1981). These spacecraft had

perigees in the 300–400 km range. A few measurements have also been obtained onboard sounding rockets (Grebowsky and





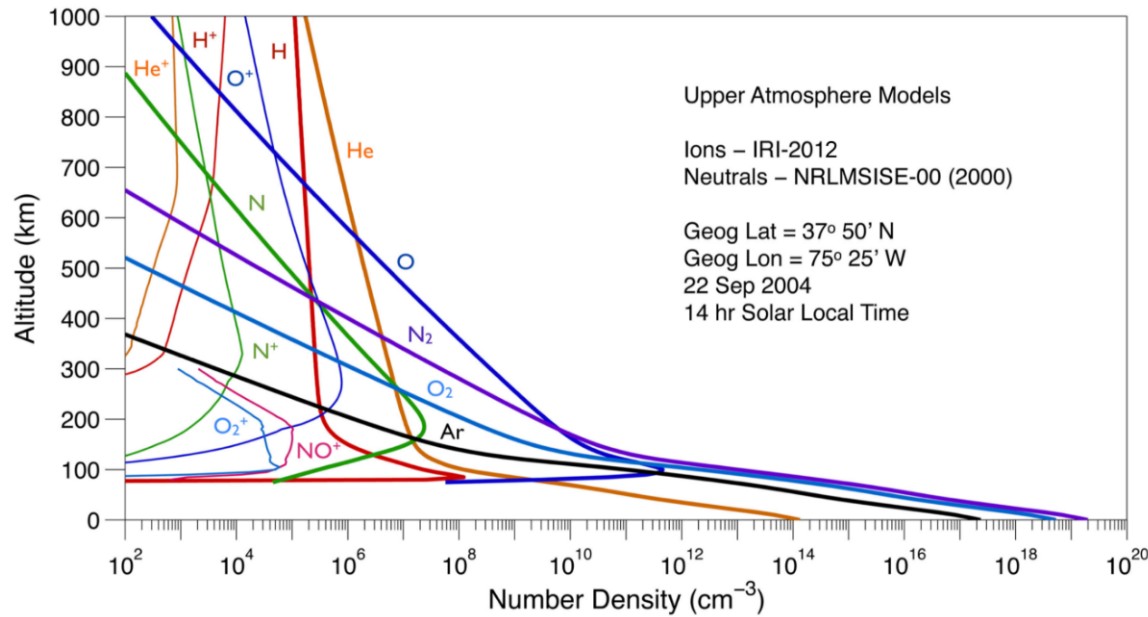

**Figure 4.** Typical example of the density altitude profiles of the major ion and neutral species obtained from the NRLMSISE-00 and IRI-2012 models. Compare with Figure 2c.

Bilitza, 2000). Ion and neutral mass spectrometry technique has been systematically used also for the study of other planetary upper atmospheres in our solar system (Waite et al., 2004; Balsiger et al., 2007; Wurz et al., 2012; Mahaffy et al., 2015). However, after Dynamics Explorer-2 mission in 1983 no other successful neutral mass spectrometer measurements have been obtained in the terrestrial thermosphere (Dandouras et al., 2018; Sarris et al., 2020).

For selected ion or neutral species, densities can be obtained also by remote-sensing optical measurements (e.g., Emmert et al., 2012; Qin and Waldrop, 2016). The NASA GOLD (Global-scale Observations of the Limb and Disk) mission, launched in 2018, consists of a UV imaging spectrograph on a geostationary satellite providing remotely measured densities and temperatures in the Earth's thermosphere for O and $N_2$ (https://gold.cs.ucf.edu/). Similarly, the NASA ICON (Ionospheric Connection Explorer) mission, launched in October 2019, includes an EUV and an FUV imager pointing at the Earth's limb
(http://icon.ssl.berkeley.edu/).

    ISR measurements allow in theory to infer the ion composition in the ionosphere, as the ISR spectra depend on the mean ion mass. However, this proves very difficult in practice (Kofman, 2000), and the ion composition is generally assumed when analysing ISR data. On the other hand, assuming an incorrect ion composition when analysing ISR data can lead to large errors in the retrieved parameters (in particular the ion temperature), which is why in several studies the assumed ion composition was
corrected using simulations from numerical models (e.g., Blelly et al., 2010; Pitout et al., 2013). A few studies have besides





made use of ISR observations to estimate the densities of some major neutral species, such as atomic oxygen and hydrogen (Blelly et al., 1992).

The scarcity of composition measurements at Earth's LTI region is thus replaced, to a certain extent, by numerical upper atmosphere models. The NCAR Whole Atmosphere Community Climate Model with thermosphere and ionosphere extension

(WACCM-X) simulates the entire atmosphere and thermospheric ionosphere, from the Earth's surface up to ~700 km altitude, and reproduces thermospheric composition, density, and temperatures in good correspondence with measurements and empirical models (Liu et al., 2018a). Besides WACCM-X, the IRAP plasmasphere-ionosphere model (IPIM) describes the transport of the multispecies ionospheric plasma from one hemisphere to the other along convecting and corotating magnetic field lines, taking into account source processes at low altitudes such as photoproduction, chemistry, and energisation (Mar-

chaudon and Blelly, 2015). It is particularly suited to the study of the $E$ and $F$ regions. $D$-region studies require a model taking into account ion and neutral species in the mesosphere as well, including cluster ions and negatively charged ions. The recently developed WACCM-D (Verronen et al., 2016) combines photoionisation by solar ultraviolet and X-ray radiation, ionization by particle precipitation and galactic cosmic rays, and a detailed chemistry scheme of 307 reactions of 20 positive ions and 21 negative ions. Particularly aimed for particle precipitation studies, WACCM-D allows for simulations of $NO_x$ production in

the mesosphere–lower-thermosphere–ionosphere (MLTI), dynamical connections to the stratosphere, and the impact on ozone (Andersson et al., 2016; Kyrölä et al., 2018; Verronen et al., 2020). The Sodankylä Ion-neutral Chemistry (SIC) model is another $D$-region photochemical model which has been used in studies of various phenomena in the MLTI (e.g., Verronen et al., 2005; Kero et al., 2008; Seppälä et al., 2018).

### 3.3.2 Neutral and ion densities

Neutral densities can be derived from a number of observation techniques. Tracking the orbital decay of space objects from the ground is one of the first techniques still applied today (Storz et al., 2005; Doornbos et al., 2008; Bruinsma, 2015). It is based on the orbit decay induced by the atmospheric drag. While tracking data are available from the 1960s onwards, their resolution ranges from one orbit to several days. A more accurate observation technique is GNSS tracking of satellites, which can provide a resolution along the orbit of up to 10 minutes, depending on the tracking accuracy and the altitude. As opposed to

tracking techniques, accelerometers provide instantaneous measurements of the non-gravitational acceleration. The first multi-year accelerometer measurements were performed by the Atmospheric Explorer missions and the Castor satellite in the 1970s (Beaussier et al., 1977).

A new era began in the year 2000 with the launch of the Challenging Minisatellite Payload (CHAMP) satellite, which carried a precise 3-axis accelerometer, star cameras and a GPS receiver as part of the scientific payload. The combination of the GPS

tracking and the accelerometer measurements allowed to obtain well-calibrated accelerations that could be used to derive accurate neutral density data at a high resolution along the orbit. The same combination of observation techniques is employed by the GRACE, Gravity Field and Steady-State Ocean Circulation Explorer (GOCE), Swarm and GRACE-FO satellites, which were launched in 2002, 2009, 2013 and 2018, respectively. All of these satellites have provided a wealth of neutral density observations in the altitude range from 200 km to 500 km.





Deriving neutral density from acceleration measurements requires knowledge of the neutral composition of the atmosphere
to accurately model the gas-surface interactions that influence the aerodynamic coefficients of the satellites. That knowledge
is based on neutral mass spectrometer data collected in the 1960s, 1970s and 1980s. As indicated in Sect. 3.3.1, since the end
of the Dynamics Explorer-2 mission in 1983, no successful neutral mass spectrometer measurements have been obtained. Like
accelerometers, neutral mass spectrometers need to be calibrated to transform the precise relative composition measurements

into accurate absolute number densities. The derivation of the neutral density and wind from the accelerometer observations,
when the accelerometer is located in the centre-of-mass of the satellite, is based on the measurement of the total linear non-
gravitational acceleration by the instrument. For a 3-axis accelerometer, the raw accelerometer observation vector $\mathbf{a}_{\mathrm{obs}}$ typically
needs to be calibrated by applying a 3×3 diagonal scale factor matrix $\mathbf{S}$ and by adding a bias vector $\mathbf{b}$ (Doornbos, 2011):

$$\mathbf{a}_{\mathrm{cal}} = \mathbf{S}\mathbf{a}_{\mathrm{obs}} + \mathbf{b} \tag{2}$$

Typically, accelerometer scale factors are considered to be nearly constant (Tapley et al., 2007), whereas biases are typically
estimated on a daily basis. Both the scale factors and biases can be estimated precisely from tracking by the Global Positioning
System (GPS) (Helleputte and Visser, 2009). It is anticipated that spaceborne multi-GNSS receivers will make this estimation
even more robust and precise.

    The calibrated accelerometer observations $\mathbf{a}_{\mathrm{cal}}$ include the aerodynamic accelerations $\mathbf{a}_{\mathrm{aero}}$, but also need to be reduced first

by removing other contributions:

$$\mathbf{a}_{\mathrm{aero}} = \mathbf{a}_{\mathrm{cal}} - \mathbf{a}_{\mathrm{srp}} - \mathbf{a}_{\mathrm{alb}} - \mathbf{a}_{\mathrm{IR}} - \mathbf{a}_{\mathrm{rem}}, \tag{3}$$

where $\mathbf{a}_{\mathrm{srp}}$, $\mathbf{a}_{\mathrm{alb}}$, $\mathbf{a}_{\mathrm{IR}}$ represent the accelerations caused by solar radiation pressure, Earth albedo and Earth infrared radia-
tion, respectively. The remaining accelerations $\mathbf{a}_{\mathrm{rem}}$ are assumed to be negligible. The aerodynamic acceleration is typically
modelled as (Doornbos, 2011):

$$\mathbf{a}_{\mathrm{aero}} = \mathbf{C}_a \frac{A_{\mathrm{ref}}}{m} \frac{1}{2} \rho {v_r}^2 \tag{4}$$

where $\mathbf{C}_a$ are dimensionless force coefficients (Anderson, 2010), $A_{\mathrm{ref}}$ represents a reference area, $m$ the satellite mass, $\rho$
the neutral density and $\mathbf{v_r}$ the velocity of the atmosphere relative to the spacecraft body. This velocity includes the neutral
wind. Doornbos (2011) proposed and implemented an iterative scheme for successfully deriving neutral density and wind from
accelerometer observations for low-flying satellites such as CHAMP and GOCE.

The neutral density can also be derived by adding the number densities of the individual species composing the neutral
atmosphere as measured by a neutral (or neutral and ion) mass spectrometer. This technique has been systematically used for
the study of planetary upper atmospheres (Waite et al., 2004; Balsiger et al., 2007; Wurz et al., 2012; Mahaffy et al., 2015).
Similarly, the thermal ion density can be derived by adding the number densities of the individual ion species composing the
ionosphere (Hoffman et al., 1974; Chappell, 1988; Welling et al., 2015). A low-Earth-orbiting satellite mission comprising

well-calibrated instruments such as a GPS receiver, an accelerometer, and a neutral and ion mass spectrometer could allow for
the first time to measure simultaneously neutral and ion densities and compositions to determine the accuracy of the summing
method.

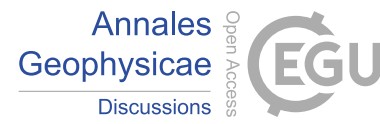

## 3.4 Neutral winds

In the LTI, neutral winds are strongly influenced by many external drivers like geomagnetic and solar activity and tidal and
gravity waves (Rees, 1989). Figures 1b and 2b show the global distribution and selected altitude profiles, respectively, of the
neutral winds during the St Patrick Day geomagnetic storm on 17 March 2015, obtained from a WACCM-X simulation. These
figures show in particular that large magnitudes of several hundred metres-per-second can be reached at polar latitudes.

The characterisation of neutral winds across a wide range of altitudes is critical to correctly quantify processes such as Joule
heating (e.g., Kosch et al., 2011) or $F$-region dynamics (e.g., Billett et al., 2020). In the lower altitude range of the LTI, the
neutral wind characteristics are poorly known. At higher altitudes, thermospheric neutral winds have been in the last decades
retrieved by, e.g., accelerometers (Doornbos, 2011) onboard many satellite missions like Dynamics Explorer, CHAMP, GOCE,
and Upper Atmosphere Research Satellite embedding a Wind Imaging Interferometer (UARS/WINDII). The accelerometer
data can be further processed with a high-fidelity geometry and aerodynamic modelling to obtain thermospheric products
(March et al., 2019a, b). The availability of cross-track accelerations has led to a large amount of horizontal cross-wind data
(Sutton et al., 2005; Cheng et al., 2008; Doornbos et al., 2010), while the vertical acceleration was generally assumed too small
to obtain reliable wind measurements (Visser et al., 2019). Vertical winds are more difficult to retrieve; however with the help
of linear and angular accelerations this was recently done with the latest release of the GOCE thermospheric data, which are
available on the ESA GOCE virtual archive (http://eo-virtual-archive1.esa.int/GOCE-Thermosphere.html).

Besides *in situ* measurements by spacecraft, various ground-based instruments enable neutral wind observations by remote
sensing. Wide-field Fabry-Perot interferometers, or scanning Doppler imagers (SDIs), measuring the Doppler shift of the
airglow/auroral red (630.0 nm) and/or green (557.7 nm) emission lines allow to retrieve the $F$-region and $E$-region neutral
winds. One example of SDI is SCANDI (Aruliah et al., 2010), which observes the red line to measure neutral winds at around
250 km altitude within a large field of view in multiple horizontal bins giving a spatial resolution of the order of 100–300 km,
with a time resolution of about 8 min. Narrow-field Fabry-Perot interferometers (NFPIs) use the same principle to observe
neutral winds within smaller spatial bins ($< 10$ km) with a high precision in the pointing direction (Shiokawa et al., 2012).
The downside of those ground-based optical instruments is that they require clear and dark skies to provide neutral wind
measurements. A cross comparison of SDI and NFPI measurements can be found in Dhadly et al. (2015). Finally, incoherent
scatter radars can also allow to estimate neutral winds using a method called stochastic inversion (Nygrén et al., 2011). While
they provide a lower time resolution and larger uncertainties, on the other hand they allow to retrieve altitude profiles in the $E$
region (95–135 km altitude in 10 km bins) and are not affected by cloud cover nor daylight.

Various empirical models of neutral winds have been built by combining large data sets consisting of observations from satel-
lites, rockets and ground-based instruments. The prime example of neutral wind climatologies is the Horizontal Wind Model
(HWM) series (Drob et al., 2008, 2015). The HWM model is constantly under development at the Naval Research Lab and its
latest edition is the HWM-14 (Drob et al., 2015). Neutral winds are also studied using first-principle models, wherein equa-
tions describing dynamics, as well as photochemical, transport, electrodynamical, thermodynamical and radiative processes are
solved self-consistently. Examples of such models include, e.g., the Thermosphere Ionosphere Electrodynamics General Cir-





culation Model (TIE-GCM; Richmond et al., 1992), WACCM (Liu et al., 2010), the Global Ionosphere Thermosphere Model (GITM; Ridley et al., 2006), and the Magnetosphere-Thermosphere-Ionosphere Electrodynamics General Circulation Model (MTIE-GCM; Peymirat et al., 1998).

### 3.5 Ion drift speed and Electric fields

Ionospheric convection corresponds to the plasma drift relative to the neutral medium, being typically from dayside to night-side through the midnight meridian and back towards the dayside at auroral latitudes. Convection is an important ionospheric parameter which reflects the complex coupling between the solar wind and the magnetosphere as well as internal magneto-spheric processes such as reconnection in the magnetotail (Dungey, 1961; Cowley and Lockwood, 1992). The high-latitude flows generally form two cells, with anti-sunward flow over the polar cap and return sunward flows at lower latitudes in the au-roral zones, both in the evening and morning sectors. However, both the spatial extent of the flow system and the magnitude of the flows vary and are related to the solar wind parameters, in specific to the north-south ($B_z$) and east-west ($B_y$) components of the interplanetary magnetic field (IMF, e.g., Thomas and Shepherd, 2018).

Ionospheric ion drifts commonly refer to the $F$ region above 200 km, where collisions between ions and neutrals are scarce, and the relationship between ion velocity $\mathbf{v}_i$ and electric field $\mathbf{E}$ is given by $\mathbf{v}_i = \mathbf{E} \times \mathbf{B}/B^2$, where $\mathbf{B}$ is the Earth's magnetic field and $B$ its magnitude. Therefore, strong ion flows correspond to strong electric fields. This is illustrated in the global distribution and example profiles of the ion drift speed given in Fig. 1c and 2b, obtained from a WACCM-X simulation of the St Patrick's Day storm and revealing that ion drifts take place at high latitudes only, where strong electric fields are present.

Because magnetic field lines are equipotentials due to high parallel conductivities, plasma convection is almost perfectly projected from the magnetosphere to the ionosphere (e.g., Weimer et al., 1985; Marchaudon et al., 2004), when taking into account the magnetic field convergence towards the surface, as can be seen in Fig. 2b through the fact that ion drift speeds do not exhibit a significant altitude variability above ∼150 km. However, ionospheric convection, or ion drift, displays rapid variations of the order of a few minutes, which directly reflect the variable solar wind-magnetosphere coupling. Ion drift measurements have been developed in the 1970s-80s with the building of ground-based facilities (high-frequency (HF) coherent radars, ionosondes and incoherent scatter radars) and the launch of satellites flying in the ionosphere or at higher altitudes. For clarity, this section is divided into subsections reviewing the different techniques.

### 3.5.1 HF coherent radars and SuperDARN

HF coherent radars transmit oblique waves and use ionospheric refraction of the signal to reach very large distances from the radar. They generally cover a large field-of-view (50–60°) by sounding several successive beams with a high-temporal resolu-tion (1–2 min). A backscattered signal on ionospheric density irregularities aligned with the magnetic field allows retrieving the Doppler shift of the echoes, giving access to the ion drift at successive distances along the beam (15–45 km resolution) (Greenwald et al., 1985; Villain et al., 1985). The main limitation of this measurement technique is the access to only one com-ponent of the ion drift, called line-of-sight velocity. To overcome this, HF radars have been paired such as to have a common volume of sounding where full ion horizontal velocity vectors can be reconstructed.





At the beginning of the 1990s, a network of HF radars called SuperDARN was developed and pairs of radars were built first along the auroral zones of the northern and southern hemispheres in order to reconstruct the global convection pattern at high latitudes (Greenwald et al., 1995). However, the often incomplete coverage of measured echoes in the field of view of each radar did not allow good combined measurements coming from pairs of radars. To overcome this problem, statistical maps of convection were first built from one radar, gridded in geomagnetic latitude and MLT and binned with respect to IMF

(Ruohoniemi and Greenwald, 1996; Ruohoniemi and Baker, 1998). These maps were then combined with real line-of-sight velocities of each radar to enhance the realistic representation of the ion drift. This method allows a continuity of coverage in each polar hemisphere, but is only fully representative in regions where real fitted vectors are reconstructed from radars measurements. More recently, the radar network has been extended in the polar cap and mid-latitude regions (Nishitani et al., 2019) allowing for a better coverage during perturbed periods. New versions of statistical maps have also been proposed

using all radars in each hemisphere over a larger time period and with different types of binning to take into account seasonal variations, mid-latitudes echoes, IMF variations or directly geomagnetic activity levels (Ruohoniemi and Greenwald, 2005; Pettigrew et al., 2010; Cousins and Shepherd, 2010; Thomas and Shepherd, 2018).

The strength of the SuperDARN radars is their capability to follow large-scale and meso-scale convection with an excellent spatio-temporal resolution, whose patterns are often similar to the statistical maps obtained for equivalent IMF and/or tail

conditions (e.g., Provan et al., 1999; Wild et al., 2003; Huang et al., 2000; Senior et al., 2002; Imber et al., 2006). Complete reviews of SuperDARN radars can be found in Chisham et al. (2007) and Nishitani et al. (2019).

### 3.5.2 Ion drift measurements onboard satellites

Ion drift has also been systematically recorded with satellites, originally through the combination of ion-drift metre (IDM) and retarding potential analyser (RPA) measurements. The IDM measures velocities in the direction perpendicular to the satellite

velocity vector; the RPA measures along-track velocities, and provides estimates of ion composition and ion temperature as well. The first satellites to make such measurements were AE-C in the 1970s (Hanson et al., 1973) and DE-2 in the 1980s (Heelis et al., 1981). To this day the AE-C measurements remain the only published satellite-based measurements of the LTI below 200 km altitude. The AE-C observations were followed by the series of DMSP satellites, which operate around 800 km altitude on polar orbits with orbital periods spanning from 90 to 120 min. The DMSP spacecraft generally cross the auroral

zone and the polar cap region close to the dawn-dusk plane in about 10 to 15 min, which makes a full reconstruction of the instantaneous convection pattern impossible. However, these different satellites have been used to study specific localised phenomena with great success, such as the convection in the polar cusp, cleft and cap (e.g., Heelis et al., 1976; Heelis, 1984; Burch et al., 1985; Heelis et al., 1986), and the properties of enhanced convection at subauroral latitudes known as subauroral ion drift (SAID) or subauroral polarisation stream (SAPS) (e.g., Spiro et al., 1979; Anderson et al., 1991). Moreover, the DMSP

data have successfully been used to build statistics of high-latitude convection patterns with respect to IMF conditions, seasons and hemispheres, from which well-known models have been derived (e.g., Heelis et al. (1982) and Hairston and Heelis (1990) with DE-2 data, or Rich and Hairston (1994) and Papitashvili and Rich (2002) with DMSP data).





The Swarm mission (Friis-Christensen et al., 2008), launched in 2013, consists of three satellites in polar circular orbits at altitudes of ∼450–500 km. Swarm incorporated a new method of measuring ion drift and temperature known as thermal ion imaging, or TII (Knudsen et al., 2017). TII sensors produce 2-D images of the low-energy ion distribution at rates as high as 125 s$^{-1}$, and can be used to determine higher-order features of the ion distribution functions such as ion temperature anisotropy (Archer et al., 2015). Lomidze et al. (2019) showed that the Swarm TII cross-track ion velocity measurements are consistent with a DE-2-based convection model (Weimer, 2005). TII measurements from Swarm have revealed the existence of intense flow channels at the boundary of the nightside R1/R2 FAC systems (Archer et al., 2017), and in association with sub-auroral "STEVE" arcs in which the ion flow velocity can exceed 5 km/s (MacDonald et al., 2018). Swarm ion flow measurements have been used in conjunction with magnetic field measurements in a number of studies of low-frequency electrodynamics including measurements of quasi-static Poynting flux (see Sect. 4.3), and magnetosphere–ionosphere–thermosphere (MIT) coupling via Alfvén waves (Park et al., 2017a; Miles et al., 2018; Pakhotin et al., 2018, 2020).

### 3.5.3 Incoherent Scatter Radars

Other ground-based instruments such as ISRs can also yield the ion drift (e.g., Caudal and Blanc, 1983; Rishbeth and Williams, 1985) and dynasondes (Wright and Pitteway, 1982). ISRs measure the spectrum of ion acoustic waves, which gives information on several plasma parameters, including one component of the ion velocity. By either pointing the radar beam to nearby positions in a cycle or by using additional receivers (the tri-static capability of the European Incoherent Scatter (EISCAT) radar), one can get the full 3D ion velocity vector, from which the electric field in the *F* region can be derived. The advantage is that spatially small-scale features with relatively high time resolution (typically from a few tens of seconds to a few minutes) can be studied, but the measurements cover only a localised volume. However, by changing the beam elevation and azimuth, larger latitudinal coverage can also be obtained. Those measurement modes have been used to build empirical models at low, middle and high latitudes (e.g., Richmond et al., 1980; Foster, 1983; Holt et al., 1987; Senior et al., 1990).

The small-scale variability in electric fields at high latitudes is typically related to the electrodynamics of auroral arcs or magnetospheric processes during substorms. By using the EISCAT ISR measurements, it has been established that auroral arcs are often associated with narrow intense electric fields just outside of the auroral arcs and related increased electron densities due to auroral electron precipitation (Aikio et al., 2002). Cluster satellite measurements showed that those electric fields develop rapidly in a time scale of minutes (Marklund et al., 2001; Aikio et al., 2004). Additionally, intense flow channels of ionospheric plasma have been found on the dayside in the cusp region (Oksavik et al., 2004), in the polar cap (Nishimura et al., 2014), and at high latitudes on the nightside in association with magnetospheric bursty bulk flows (Pitkänen et al., 2013).

ISR measurements are also ideal for extreme velocities. Aikio et al. (2018) reported extremely high ion speeds reaching over 3000 m s$^{-1}$ (about ten times higher than the normal convection velocities) and verified the observation by three independent measurements, the EISCAT UHF and VHF radar electric field and ion temperature measurements as well as the Swarm satellite Electric Field Instruments (EFI). Aikio et al. (2018) suggested that the observed flow channel accommodates increased nightside plasma flows during the substorm expansion phase as a result of reconnection in the near-Earth magnetotail. These narrow regions of high ion speeds facilitate strong ion-neutral frictional heating. No global convection models produce these





features. The next generation EISCAT_3D ISR will be able to address this small-scale variability by conducting volumetric measurements of plasma parameters including vector ion drifts in Northern Fennoscandia starting from 2022 (McCrea et al., 2015).

### 3.5.4 Numerical simulations

Several global models simulate the plasma convection around the Earth and the consecutive ion drift pattern within the ionosphere. The system can be modelled based on first principles, e.g., using a magnetohydrodynamic (MHD) model (e.g., Wiltberger et al., 2004; Honkonen et al., 2013; Gordeev et al., 2015). In this approach, the general plasma circulation within the magnetosphere is mapped into the ionosphere where it is used to determine the ionospheric electric field and the plasma drift pattern (e.g., Janhunen et al., 2012). While all global MHD simulations provide the ionospheric plasma drift pattern and the electric field, often the resulting polar cap potential can be overestimated (e.g., Haiducek et al., 2017), or underestimated (e.g., Palmroth et al., 2005) compared to best available measurements. A more measurement-based method to model ionospheric electric fields and plasma drifts is to assimilate ion drift data coming from either SuperDARN convection maps or DMSP observations into an electrodynamics coupling model such as Assimilative Mapping of Ionospheric Electrodynamics (AMIE) technique (Richmond and Kamide, 1988; Cousins et al., 2015). SuperDARN and/or DMSP data can also feed ionospheric models such as the IRAP Plasmasphere-Ionosphere Model (IPIM; Marchaudon and Blelly, 2015; Marchaudon et al., 2018). All these modelling techniques require accurate measurements of the ionospheric parameters such as conductivities, which can be used to restrict the models to give more realistic results. This is important, because many of these models use the ionospheric solution as a boundary condition within the magnetosphere, thus emphasising the role of the ionosphere to provide forcing for above (e.g., Ridley et al., 2004).

### 3.6 Magnetic fields

Measurements of the magnetic field vector $\mathbf{B}$ provide a key parameter for studying ionospheric electrodynamics. Magnetic field variations allow to detect *in situ* and distant electric current density, $\mathbf{j}$, through Ampère's law, $\nabla \times \mathbf{B} = \mu_0 \mathbf{j}$. The largest contributions to the Earth's magnetic field are due to sources within Earth's core and crust, and from large-scale magnetospheric currents (e.g., Olsen and Stolle, 2012). After removal of these contributions (e.g., as provided by geomagnetic field models) magnetic residuals are particularly valuable for studying ionospheric currents sources (e.g., Stolle et al., 2017). Especially in target are the FACs (see Sect. 4.2) connecting the magnetosphere to the ionospheric $E$ region. At high latitudes, the magnetic residuals $\delta\mathbf{B}$ and the derived FACs, together with electric field observations are crucial to quantify significant parts of energy deposition into the upper atmosphere by magnetic forcing (Sect. 4.1), Poynting flux (Sect. 4.3) or Joule heating (Sect. 4.7).

Magnetic signatures due to ionospheric currents are of much lower amplitude at middle and low latitudes. Nonetheless, satellite-based magnetic field observations are indispensable in understanding the global distribution of currents. Special attention is given to currents which are connected to $E$-region and $F$-region dynamos. These are, for example, inter-hemispheric currents connected to midlatitude Sq currents or low-latitude $F$-region dynamo currents (e.g., Olsen, 1997; Park et al., 2020; Lühr et al., 2019), low-latitude gravity-driven and plasma-pressure-driven currents (e.g., Alken et al., 2017), variations of

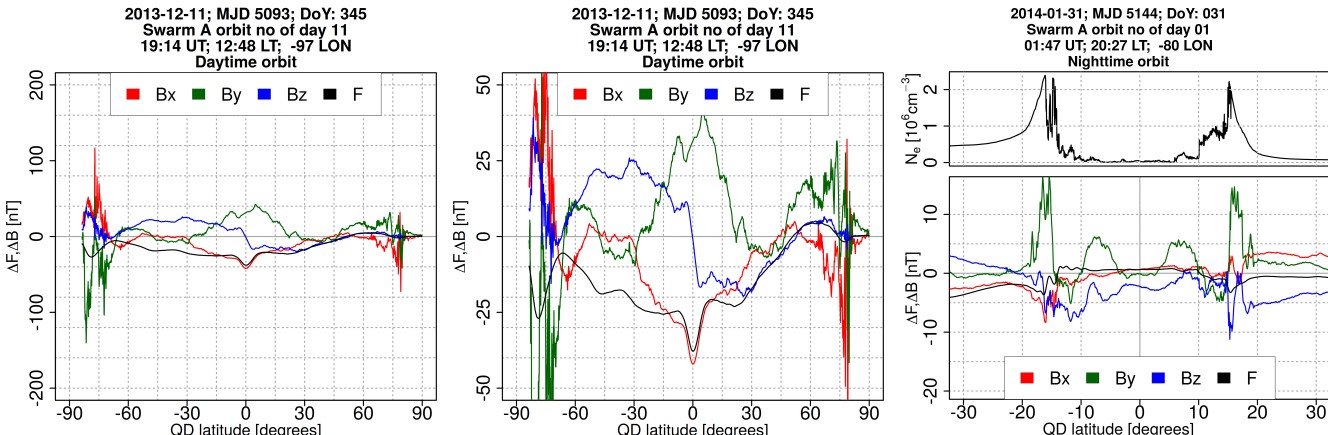

**Figure 5.** Left: a dayside orbital segment of magnetic field signatures of ionospheric currents from the *Swarm* mission. Middle: same orbital segment but zoomed in magnitude. Right: low-latitude orbital segment of a nightside orbit. The location and time of the orbital segments are provided in the panel title. The coordinate system is local and it is x-north, y-east, z-up.

equatorial electrojet currents in response to wave coupling from the middle and lower atmosphere (e.g., Yamazaki et al., 2017), or electromagnetic characterisation of post-sunset equatorial plasma irregularities (e.g., Rodríguez-Zuluaga and Stolle, 2019). Figure 5 shows two examples of Swarm spacecraft orbital segments that reflect different ionospheric currents that may be monitored with high precision magnetometers. Shown are differences between the magnetic data and predictions of the CHAOS-6 magnetic field model Finlay et al. (2016) to eliminate contributions from the core, crustal and large-scale magnetospheric field. This day was geomagnetically quiet with Kp ≤ 1 the entire day. The strongest signals arise from auroral currents, and field-aligned currents are most pronounced in the eastward y-component. At the mid- and low latitudes day-side *E*-region currents or currents associated to post-sunset plasma depletion in the *F* region dominate the signal.

High-precision magnetic field measurements such as those available by the CHAMP and Swarm satellites from altitudes between 350 km and 500 km have tremendously improved our understanding of ionospheric phenomena from high to low latitudes (e.g. Lühr et al., 2004; Alken and Maus, 2007; Alken, 2016; Park et al., 2017a; Park et al., 2020). However, similar magnetic field measurements taken at altitudes of 200 km and below, diving into the largely unexplored *E*-region dynamo, would provide an invaluable key to understand the coupling between the atmosphere, solar radiation, and the geomagnetic field through unprecedented data.

## 4 Derived LTI parameters: Current understanding

This section reviews the main LTI parameters which are not observed directly but rather derived from measurements of the parameters discussed in Sect. 3. For each of them, their current understanding and characterisation and their description in numerical models are discussed. The derived parameters are ordered in this section starting from those associated with forcing





of the LTI from above and moving on to those which characterise local properties of the LTI, and finally covering the forcing from below.

## 4.1 Magnetic forcing and general energy circulation

Forcing of the upper atmosphere is achieved essentially by energy and momentum transfer between charged and neutral particles. Energy deposition drives collisional heating of the neutrals, whereas the momentum imparted to the plasma by the Lorentz force, $\mathbf{j} \times \mathbf{B}$, is exchanged with the neutrals by collisional friction. The subject is a part of a broader concept called energy circulation or energy transfer. At polar latitudes, both energy and momentum are primarily extracted from the solar wind (Axford and Hines, 1961; Akasofu, 1981; Palmroth et al., 2003; Palmroth et al., 2006b) in a sequence of processes that include reconnection at the dayside magnetopause (e.g., Crooker, 1979; Trattner et al., 2007; Hoilijoki et al., 2014) driving magnetospheric convection (Dungey, 1961). The phenomena are mediated by magnetic field lines, which are essential also for FACs (Sect. 4.2) that transfer momentum, and for the Poynting flux (Sect. 4.3) that transfers energy. Even though the strict meaning of *magnetic forcing* is related to the momentum transfer and Lorentz forces, the term often covers (also) the effects of energy transfer and Joule heating. For example, one of the goals of the Swarm mission (Friis-Christensen et al., 2006), namely *quantification of magnetic forcing of the upper atmosphere*, addresses variations in the neutral density and atmosperic upwelling in response to Joule heating (e.g., Lühr et al., 2004; Prölss, 2011). In this section, we discuss magnetic forcing mainly as the transfer of momentum via Lorentz forces, whereas Joule heating and heat transfer to the neutrals are discussed in sections 4.7 and 4.8, respectively.

Since FACs are force-free ($\mathbf{j} \parallel \mathbf{B}$ hence $\mathbf{j} \times \mathbf{B} = 0$), they are ideal transmitters of momentum between the high and low altitude ends of the MIT system, as part of the auroral current circuit (Boström, 1964). Lorentz forcing in the equatorial magnetosphere, due to currents flowing across the magnetic field, is transferred by FAC tangential stress to Lorentz forcing in the ionosphere due to the horizontal current that closes the FAC (Iijima, 2000). The capability of the ionosphere to carry an electric current is a strong constraint for the seed region of the magnetic forcing, limiting it to a rather narrow range, essentially the ionospheric $E$-layer, at ∼90–150 km altitude. In the lower part of this layer, the electrons become collisionless and their $\mathbf{E} \times \mathbf{B}$ drift provides the Hall component of the ionospheric current, $\mathbf{j}_H = -\sigma_\mathrm{H} \mathbf{E} \times \mathbf{B}/B$, perpendicular to the electric field, with $\sigma_\mathrm{H}$ the Hall conductivity. The ions become collisionless just at the upper side of the $E$-layer and their motion along the electric field provides the Pedersen current, $\mathbf{j}_P = \sigma_\mathrm{P} \mathbf{E}$, with $\sigma_\mathrm{P}$ the Pedersen conductivity (see Sect. 4.6). Above the $E$-layer, electrons and ions drift together and the ionospheric current vanishes.

While Joule heating is related to the Pedersen current, the Lorentz force has contributions from the total ionospheric current, i.e., the sum of the Pedersen and Hall currents. Another way of writing the total current is to express it as the sum of a curl-free and a divergence-free component (e.g., Vanhamäki and Juusola, 2018). The magnetic forcing is associated directly with the curl-free component, that actually closes the FAC. The divergence-free component is not subject to local magnetic forcing exerted by FAC closure, although it plays a key role in the inductive storage and release of energy, during transient MIT coupling (Yoshikawa, 2002a, b). While such transients are essential for MIT dynamics, their time scales are typically short, between a few seconds and a few minutes, depending on the related spatial scales (Yoshikawa, 2002b). For the rest of the time,





under quasi-static approximation, the curl-free component of the ionospheric current is dominated by the Pedersen current, while the Hall current is largely divergence-free (Vanhamäki et al., 2012), therefore the Pedersen current appears to be the main agent of magnetic forcing, be it related to Joule heating or the Lorentz force.

A simple example is provided by the quiet auroral arc (or oval, on a larger scale), which can be approximated with an ideal, 1D structure, where the electric field is normal to the arc, the FAC is closed horizontally across the arc by the Pedersen current, while the Hall current along the arc is divergence-free (e.g., Marghitu, 2012). In this case, the Lorentz force on the Pedersen current is directed along the arc (or auroral zone) and drives plasma convection against the collisional drag of the neutrals. At the same time, the Poynting flux carried by the FAC is in balance with the Joule heating of the Pedersen current that closes the
FAC and energy dissipation is essentially local, i.e., limited to the FAC closure area (Richmond, 2010; Vanhamäki et al., 2012). Not surprisingly, magnetic forcing is more intense during increased geomagnetic activity, when the FAC, particle precipitation, conductivities, and electric field can be highly variable and non-uniform. At such times, smaller-scale enhancements can make a significant contribution to the forcing (Codrescu et al., 1995). The relationship between driver and response is also considerably more complex. For example, the Hall current may also contribute to the FAC closure, and dissipation can become non-local
(Fujii et al., 2011).

The magnetic forcing described so far applies to the cases where the ionosphere–thermosphere system behaves like a load, $\mathbf{E} \cdot \mathbf{j} > 0$, controlled by magnetospheric processes. The energy and momentum influx carried by FAC is thermally dissipated (analogous to a resistance in an electric circuit), and can also feed the coherent convection of neutral wind (analogous to a motor). Since the neutral atmosphere dominates the ionosphere at $E$-layer heights, its inertia is comparatively large and the
latter can become important only in the case of intense events whose duration is long enough, typically strong storms or substorms. In such cases, the opposite, flywheel effect (e.g., Deng et al., 1991, 1993; Paschmann et al., 2003), is observed as well during the recovery stage, when the forcing on the ionosphere stops (or decreases) whereas the neutral atmosphere needs a longer time to come to rest. At such times, the neutral wind is analogous to a dynamo and the ionosphere–thermosphere system behaves like a generator, $\mathbf{E} \cdot \mathbf{j} < 0$, playing an active role in the interaction with the magnetosphere.

## 4.2   Field-aligned currents

Field-aligned currents (FACs) were first suggested to connect the upper ionised atmosphere to the outer space by Birkeland (1908), and their existence was confirmed some 60 years later by satellite observations (Zmuda et al., 1966; Cummings and Dessler, 1967). Ever since, FACs have been one of the most central topics in space plasma research. Satellite observations of magnetic field variations $\delta\mathbf{B}$ are often used to determine the *in situ* FAC density $j_\parallel$ (e.g., Iijima and Potemra, 1978; Ritter
et al., 2013; Lühr et al., 2015; McGranaghan et al., 2016). The FAC density is often expressed using the infinite current sheet approximation:

$$j_\parallel = \frac{1}{\mu_0} \frac{\partial B_x}{\partial t} \frac{1}{v_n} \tag{5}$$

where the Cartesian reference frame is such that $x$ is parallel to the sheet direction (direction of maximum variance), $y$ is normal to the sheet plane (direction of intermediate variance), and $z$ is along the mean magnetic field, while $v_n$ is the projection of





the spacecraft velocity along the $y$-axis (Marchaudon et al., 2006). The three-dimensional current system can be derived from decomposition analysis of all vector components (e.g., Laundal et al., 2018). Of various topologies and scale sizes, FACs connect together magnetospheric regions having different controlling parameters, like the plasma sheet or the low-latitude boundary layer, to the auroral zone in the LTI. They also play an essential role in the magnetic forcing of the upper atmosphere (see Sect. 4.1). FAC structures, consisting of upward and downward currents with planar or filamentary geometry (Boström,

1964), confine the transport of Poynting and momentum fluxes inside the current system, on large, meso-, and small scales. Reviews addressing the morphology and physics of FACs are available, e.g., in the AGU Geophysical Monographs edited by Ohtani et al. (2000) and Keiling et al. (2018).

      Figure 6 shows the typical large-scale high-latitude current system including FACs and horizontal currents, with the polar cap convection cells under southward IMF driving shown with thin black lines. The most common FAC structures are the

so-called Region 1 (R1) and Region 2 (R2) currents (Iijima and Potemra, 1976). The R1 (poleward) / R2 (equatorward) current system consists of two thick current sheets, with the R1 current pair flowing downwards (in blue in the figure) and upwards (in red) in the dawn and dusk sectors, respectively, while the R2 system flows in the opposite direction. Horizontal currents in the high-latitude LTI consist of Hall (in orange) and Pedersen (in green) currents associated with the FACs. Large-scale eastward/westward electrojets are associated with the convective electron motion on the dusk and dawn sides, respectively (e.g.,

Baumjohann, 1983). The midnight sector current distribution in Fig. 6 depicts the substorm current wedge (McPherron et al., 1973; Birn et al., 1999; Keiling et al., 2009) consisting of thick downward/upward current filaments on the dawn/dusk side, connected in the ionosphere by the substorm (westward) electrojet. An additional FAC distribution (not shown in Fig. 6), called NBZ (northward IMF $B_z$) or R0 current system, appears within the noon sector during strongly positive IMF $B_z$ orientation and manifests reconnection at the tail lobes and consequent sunward plasma flows. While Iijima and Potemra (1976, 1978)

needed several months of data to infer the large-scale FAC patterns, at present, the AMPERE network of *in situ* engineering magnetometers, operational on the Iridium satellites, is able to follow the large-scale FACs almost in real time (Anderson et al. (2014); recent reviews by Milan et al. (2017) and Coxon et al. (2018)).

      FAC current systems are also observed at smaller scales, of which auroral arcs provide the best example (Partamies et al., 2010). A recent review is provided by Karlsson et al. (2020). Within this example, the upward FAC sheet (consisting mainly

of precipitating electrons) above the arc is paired with a downward FAC sheet (e.g., Elphic et al., 1998), similar to the scaled down R1/R2 system. Multiple arcs can consist of multiple pairs of upward and downward FACs, but are also observed to share a unique current system, with all the arcs on the upward FAC leg (Wu et al., 2017). Another example of scaled-down current system, similar to the substorm current wedge, is provided by the current circuit that connects magnetospheric bursty bulk flows (Baumjohann et al., 1990; Angelopoulos et al., 1992) with ionospheric north–south auroral structures (Henderson et al.,

1998), also known as streamers. It has also been suggested that the substorm current wedge consists of several such wedgelets (Liu et al. (2015); recent review by Liu et al. (2018b)).

      Planar and filamentary FAC structures, connected by meridional and zonal ionospheric current, respectively, correspond to the two basic configurations of the auroral current circuit anticipated by Boström (1964), illustrated above with specific examples. Mixed configurations are observed as well; for instance, event studies (Marghitu et al., 2009, 2011) and statistical





**Figure 6.** Schematic view of the high-latitude ionospheric current system, showing the configuration of the driving field-aligned currents of region 1 (along field lines closing via the outer magnetosphere) and region 2 (closing via the opposite hemisphere). Also shown schematically are the locations and configuration of the ionospheric convection pattern, Pedersen and Hall currents, the substorm current wedge and auroral electrojets.

evidence (Jiang et al., 2015) suggest that a current system consisting of planar FAC sheets and ionospheric FAC closure in the zonal direction may develop during the substorm growth phase. A different kind of mixed configuration is observed for active Alfvénic arcs, where the planar FAC sheet can break into current filaments (Chaston et al., 2011). While sheets and filaments provide simple geometries, convenient to organise FACs according to their scales, actual observations are rarely clear-cut and reflect superpositions of geometries and scales, often more complex also than the two above examples of mixed configurations.

Various techniques have been developed to deal with actual FAC data, able to explore their multi-scale structure (e.g., Bunescu et al., 2015), or to take advantage of multi-point *in situ* information, as provided, e.g., by the Cluster and Swarm missions (e.g., Dunlop et al., 2002; Marchaudon et al., 2009; Ritter et al., 2013; Blăgău and Vogt, 2019; Vogt et al., 2020). An alternative option, which benefits from prior development and validation with ground-based data, is the spherical elementary





current systems (SECS) technique (Amm, 1997; Amm and Viljanen, 1999), adapted also for Swarm data (Amm et al., 2015).
A key advantage of this approach is that, when used with LEO satellite data, it provides both the FACs and the ionospheric currents in a consistent manner (such that the FAC density is equal to the divergence of the curl-free ionospheric current). On the other hand, its accuracy and resolution depend on the distance above the ionospheric current, in particular for the divergence-free component (typically dominated by electrojet Hall current), whose effect is observed remotely. Statistical investigations of FAC and ionospheric currents, based on Swarm data, were published, e.g., by Lühr et al. (2016), Huang et al. (2017), and
Workayehu et al. (2019).

### 4.3 Poynting flux

The problem in assessing ionospheric energy deposition using ionospheric measurements only is that several elements need to be evaluated simultaneously so that the total dissipation can be assessed. A possible way to overcome this problem is to evaluate the total electromagnetic energy, i.e., the Poynting flux $\mathbf{S} = \mathbf{E} \times \delta\mathbf{B}/\mu_0$ towards the ionosphere both using observations and
numerical simulations. If this assessment can be carried out on high enough orbits, the assumption is that the Poynting flux includes both the energy dissipated in Joule heating as well as the energy within particle precipitation. The main caveat in this technique is to evaluate the magnetic field such that it does not contain contributions from the dipole, so that it represents the extra electromagnetic energy towards the ionosphere. Therefore the field-aligned component of the Poynting vector can be evaluated from the quasi-static electric and perturbation magnetic fields $\mathbf{E}$ and $\delta\mathbf{B}$ measured above the ionosphere. In the limit
of quasi-static planar current sheets, the Poynting flux can be shown to be equal to the Joule dissipation $\mathbf{J} \cdot \mathbf{E}$ integrated along the magnetic field line below the spacecraft (Kelley et al., 1991).

Studies of high-latitude Poynting flux have been carried out with DE-2 (Gary et al., 1994) and, more recently, Swarm (Park et al., 2017a; Pakhotin et al., 2018, 2020) missions. Rodríguez-Zuluaga et al. (2017) used Swarm to resolve magnetic-field-aligned Poynting flux at low latitudes with a resolution of the order of 1 µW/m$^2$. Waters et al. (2004) introduced a method to
characterise the high-latitude Poynting flux towards the ionosphere by combining electric field measurements of the Super Dual Auroral Radar Network (SuperDARN) with the Iridium constellation estimating magnetic perturbations. This technique has the advantage of allowing the assessment of the net electromagnetic energy transferring to the ionosphere, including the energy deposited to drive the neutral winds while not having to estimate the ionosphere conductivity. The technique presented in Waters et al. (2004) agrees with the DMSP satellite *in situ* measurement to a few mW/m2. They estimate the total electromagnetic
energy flux of the order of 50 GW, maximising in the morning and afternoon sectors.

The topic is also ideal for global numerical simulations covering the entire solar-wind–magnetosphere–ionosphere system. These simulations can be used to assess the electric and magnetic fields within large volumes and map them to the ionospheric plane. In fact, global MHD simulations have shown that the Poynting flux starts to focus towards the magnetosphere and ionosphere already from the solar wind in regions where the open field lines are dragged towards the tail (Papadopoulos et al.,
1999; Palmroth et al., 2003; Palmroth et al., 2006c). Zhang et al. (2012) used a global MHD simulation and mapped the Poynting flux in the tail to the ionosphere, and confirmed that its ionospheric distribution reproduced the global morphology of the Poynting flux measured by the Polar satellite. While these studies have been carried out using MHD simulations that





have severe limitations in reproducing kinetic plasma physics within the magnetosphere, the results are indicative of processes

that need to be rigorously measured so that our magnetosphere–ionosphere system can be understood in terms of an energy

input/output system.

## 4.4 Ion–neutral cross sections

Of the derived parameters characterising the local properties of the LTI, among the most critical ones are the ion–neutral cross

sections. Two types of ion–neutral cross sections will be briefly reviewed here: (i) the ion–neutral momentum-transfer cross

sections, and (ii) the ion–neutral chemical reaction cross sections.

Ion–neutral averaged momentum-transfer cross sections, which are denoted $\sigma_{coll,in}$ here, are needed to determine the ion–

neutral collision frequencies (see sect. 4.5). They are functions of velocity-dependent momentum-transfer cross sections and

the relative velocity between particles. Using the velocity-dependent momentum-transfer cross section expression derived by

Dalgarno et al. (1958), Banks (1966) obtained the following formula for the averaged nonresonant ion–neutral momentum-

transfer cross section

$$\sigma_{coll,in} = \frac{3\sqrt{2}}{16} \pi^{\frac{3}{2}} \left( \frac{4.88\alpha e^2}{\mu} \right)^{\frac{1}{2}} \left[ \frac{k_B T_i}{m_i} + \frac{k_B T_n}{m_n} \right]^{-\frac{1}{2}}, \tag{6}$$

with $\alpha$ the neutral gas atomic polarisability, $e$ the elementary charge, $\mu$ the ion–neutral reduced mass, $k_B$ Boltzmann's constant,

$T_i$ and $m_i$ the ion temperature and mass, respectively, and $T_n$ and $m_n$ the neutral temperature and mass, respectively. This

formula was derived under the assumption that the ion–neutral interaction is predominantly due to the polarisation force

arising from induced dipole attraction by the neutral gas, hence neglecting short-range quantum mechanical repulsion effects.

This assumption holds for temperatures below 300 K, which is the order of magnitude of the temperature in the LTI (see

sect. 3.2). At higher altitudes, however, such as in the $F$ region, the ion and neutral temperatures are well above this limit

and start diverging. The dominant interaction in the $F$ region is the charge exchange of the $O^+$ ion with atomic oxygen

O. The momentum-transfer cross section for $O^+$ and O, $\sigma_{coll,O^+-O}$ is dominated by the charge-exchange process, whereas

the contribution from polarisation can be neglected in the $F$ region (Banks, 1966). While the ion–neutral cross sections are

crucial, they have never been measured within the LTI, and laboratory measurements reproducing the conditions in the upper

atmosphere are extremely challenging (Lindsay et al., 2001). Therefore, the estimates of $\sigma_{coll,O^+-O}$ have been extrapolated

from measurements and theoretical calculations at lower temperature, which are summarised in the introduction of Joshi et al.

(2018).

The study of the ion-neutral interactions requires accurate measurements of plasma and neutral species in relevant partially

ionised media, including composition of the neutral and ion species, velocity distribution of ions and electrons, as well as

ambient energy that is characterised by electric and magnetic fields, radiation, and temperature. Since such complex environ-

ments, particularly under the influence of various electromagnetic fields and with complicated composition, temperature, and

radiation fluxes, cannot easily be reproduced in a laboratory, the only way to understand the plasma-neutral gas interactions

in space is through *in-situ* observations in various environments in space (Yamauchi et al., 2019). Particularly, observations in





low-density environments with substantial neutral particle content are needed, for example, in the upper ionosphere near the
exobase of a planet or natural satellite, in comets, or in interstellar space.

Likewise, ion–neutral reaction cross sections, denoted $\sigma_{reac,in}$ here, in the LTI are poorly known. They are also crucial as
they affect the chemical reaction rates, and hence are key parameters in upper-atmosphere models. Ion–neutral reaction cross
sections have been derived from laboratory measurements of ion–molecule reaction rate constants in drift tubes (e.g., Woo

and Wong, 1971), but those estimates suffer from two main problems: they can be energy-dependent, and simple ion-neutral
relative speed distribution approximations do not hold in regions of the high-latitude LTI where ion convection speeds are high
(St.-Maurice and Torr, 1978). In such cases, the rate coefficient $k$ of a given reaction is given by

$$k = \int\limits_0^\infty \sigma_{reac,in}(v)\,v\,f(v)\,\mathrm{d}v, \tag{7}$$

with $f(v)$ the distribution of relative speeds $v$ between the reactants. In this situation, $\sigma_{reac,in}$ can be obtained by inverting

$k$ when the laboratory relative speed distribution $f_{\mathrm{lab}}(v)$ is known (St.-Maurice and Torr, 1978). Lin and Bardsley (1977)
proposed a Monte Carlo method to derive $f_{\mathrm{lab}}(v)$ to serve this purpose. Using this methodology, the reaction cross sections for
$O^+$ with the main molecular species in the LTI ($N_2$, $O_2$ and NO) were published by Albritton et al. (1977) based on drift-tube
experiments in both helium and argon buffer gases.

### 4.5   Ion–neutral collision frequencies

The collision rates for all species in the LTI constitute a fundamental set of parameters in the coupling in the atmosphere–
ionosphere–magnetosphere system. They depend on a number of terms as shown below. Under the assumption that the ion and
neutral populations in the LTI are separate Maxwellian distribution functions, the ion–neutral collision frequency $\nu_{in}$ can be
expressed as (Banks, 1966):

$$\nu_{in} = \frac{4}{3} n_n \left( \frac{8k_B}{\pi} \right)^{\frac{1}{2}} \left( \frac{T_i}{m_i} + \frac{T_n}{m_n} \right)^{\frac{1}{2}} \sigma_{coll,in}, \tag{8}$$

with $n_n$ the neutral density, $k_B$ Boltzmann's constant, $T_i$ and $m_i$ the ion temperature and mass, respectively, $T_n$ and $m_n$
the neutral temperature and mass, respectively, and $\sigma_{coll,in}$ the ion–neutral collision cross-section (see Sect. 4.4). From equa-
tion (6), one gets in the nonresonant case

$$\nu_{in} = 2.21\pi n_n \left( \frac{\alpha e^2}{\mu} \right)^{\frac{1}{2}}, \tag{9}$$

which is an expression that has been used for space physics applications such as the analysis of incoherent scatter radar

measurements (e.g., Virtanen et al., 2014). For a given ion–neutral pair, this can be further reduced as

$$\nu_{in} = C_{in} n_n, \tag{10}$$

with $C_{in}$ a numerical coefficient whose value is given for relevant nonresonant ion–neutral pairs in the ionosphere by Schunk
and Nagy (1980, see Table 6). Resonant charge-exchange interactions occur when a neutral and an ion from the same species





collide with each other; the corresponding ion–neutral collision frequencies, which are also given in Schunk and Nagy (1980, see Table 5), depend on $T_r = (T_i + T_n)/2$.

Besides laboratory experiment extrapolation to LTI conditions, ion–neutral collision frequencies have been estimated directly in the ionosphere using incoherent scatter radars. Nygrén et al. (1989) developed a method using the EISCAT radar to determine the ion–neutral collision frequency in the $E$ region. It uses radar observations in the vertical, field-aligned and eastward-tilted directions to obtain measurements of the ion velocity and the electric field vector, which are used to infer $\nu_{in}$ from the momentum equation for the ions. This method was applied to estimate $\nu_{in}$ within 95–130 km altitude in Nygrén et al. (1989). It was used by Oyama et al. (2012) to study the temporal variations of the ion–neutral collision frequency within 106–135 km altitude during an ionospheric heating event. Alternative methods employing incoherent scatter radars to infer the ion–neutral collision frequency in the LTI can be found in Kosch et al. (2011), who combined incoherent scatter radar and optical observations to estimate $\nu_{in}$ in the vicinity of an auroral arc, as well as in Nicolls et al. (2014) who made use of multifrequency radar measurements to retrieve $\nu_{in}$ and estimate the thermospheric neutral density.

The ion–neutral collision frequency $\nu_{in}$ has been measured *in situ* only by sub-orbital rockets as they descended through the LTI (see Fig. 8 in Sangalli et al., 2009), and the measurement has never been conducted from a satellite. The rocket technique involves comparing the $\mathbf{E} \times \mathbf{B}$ drift velocity derived from the respective field measurements with direct measurements of the horizontal drift speed of ions (Watanabe et al., 1991). In the LTI, the two quantities differ by an amount that depends on the ratio $\kappa = \nu_{in}/\Omega_i$ where $\Omega_i$ is the ion gyro-frequency. Accurate determination of $\kappa$ also requires an independent measurement of the neutral wind velocity (Sangalli et al., 2009; Burchill et al., 2012). As such rocket measurements provide only a handful of observations, the ion–neutral collision frequencies remain poorly characterised, and their inferred values likely suffer from uncertainties. Collision cross-sections and collision rates are among the largest sources of errors in empirical models, General Circulation Models and magnetosphere–ionosphere coupling simulations, for which they are key inputs. Furthermore, they represent the largest source of uncertainty in estimating the ionospheric conductivities, which are key parameters in current coupled models of the LTI.

### 4.6 Ionospheric conductivities

Ionospheric conductivities are also particularly important derived local parameters, as they are required especially in modelling, and further required to understand Joule heating. The ionospheric conductivity tensor is specified with the parallel conductivity $\sigma_\parallel$ (along the magnetic field), the Pedersen conductivity $\sigma_P$ (associated with ionospheric currents perpendicular to the geomagnetic field and parallel to the electric field defined in the reference frame moving with the conductive air at the speed of ambient neutral air in the bottomside ionosphere) and the Hall conductivity $\sigma_H$ (associated with ionospheric currents perpendicular to both the electric and magnetic fields). Specifically, $\sigma_P$ and $\sigma_H$ are expressed as

$$\sigma_P = n_e e^2 \left( \frac{\nu_{in}}{m_i(\nu_{in}^2 + \Omega_i^2)} + \frac{\nu_{en}}{m_e(\nu_{en}^2 + \Omega_e^2)} \right), \tag{11}$$

$$\sigma_H = n_e e^2 \left( -\frac{\Omega_i}{m_i(\nu_{in}^2 + \Omega_i^2)} + \frac{\Omega_e}{m_e(\nu_{en}^2 + \Omega_e^2)} \right), \tag{12}$$





with $n_e$ the electron density, $e$ the elementary charge, $\nu_{in}$ the ion–neutral collision frequency (see Sect. 4.5), $\nu_{en}$ the electron–neutral collision frequency, $m_i$ and $m_e$ the ion and electron masses, respectively, and $\Omega_i$ and $\Omega_e$ the ion and electron gyrofrequencies, respectively. Due to high electrical conductivity in the direction of the magnetic field, $\sigma_\parallel$ is very large, up to tens of S m$^{-1}$ (e.g., Yamazaki and Maute, 2017). This means that the geomagnetic field lines are nearly equipotential in the ionosphere, facilitating the approximation of the ionospheric parameters in the height-integrated form. As a consequence, the Pedersen and Hall conductivities are often presented as their height-integrated forms (i.e., conductances) as

$$\Sigma_P = \int \sigma_P(h)\,dh, \tag{13}$$

$$\Sigma_H = \int \sigma_H(h)\,dh. \tag{14}$$

The Pedersen and Hall conductivities and conductances can be estimated locally using incoherent scatter radar observations. Using the measured electron density profiles and expressions for the ion–neutral and electron–neutral collision frequencies such as given in Brekke and Hall (1988), Hall and Pedersen conductivity profiles can be derived and integrated across altitudes to obtain the corresponding conductances. Using this methodology, Aikio and Selkälä (2009) used the EISCAT radar to determine local conductances during a one-month campaign, and found that the conductances are larger in the morning sector than in the evening sector. The ionospheric conductivities can also be determined from spacecraft measurements using UV- and X-ray imaging of the aurora onboard polar-orbiting spacecraft (Aksnes et al., 2002; Aksnes et al., 2005).

In local numerical models, $\sigma_P$ and $\sigma_H$ are computed according to their definitions (eq. 11–12). However, large uncertainties exist regarding the values of the collision frequencies ($\nu_{in}$ and $\nu_{en}$), and if electron density profiles are inaccurately reproduced, this can result in significant errors in the calculated conductivities. Ionospheric conductivities have been calculated using the TRANSCAR couple kinetic/fluid transport code (e.g., Lilensten et al., 1996), more recently making use of the AMIE procedure for multi-instrument data assimilation into the model (Blelly et al., 2005). In global magnetosphere–ionosphere coupling simulations, the conductivities are absolute key parameters and can have far-reaching effects if reproduced erroneously. Often the conductances consist of two parts, an empirically derived F10.7-dependent dayside part (e.g., Moen and Brekke, 1993), while the nightside conductances are often empirical formulations based on electron precipitation (e.g., Robinson et al., 1987). Even small changes in the conductances and, e.g., adding seasonal variations can have tremendous effects in the overall modelling results (Ridley, 2007), and therefore their estimation *in situ* is of great importance.

### 4.7 Heating

The largest source of heating in the LTI is the absorption of solar EUV and UV radiation with an average rate of roughly 1 mW m$^{-2}$ (Peterson et al., 2012) or globally a few hundred GW. The solar cycle modulates these values by about $\pm 50\%$ (e.g., Lean et al., 2003). Furthermore, the heat flux maximises at the subsolar point and decreases away from it, in darkness to almost zero (except for a small contribution of Lyman $\alpha$ radiation originating from the geocorona; e.g., Maeda, 1969; Waldrop and Paxton, 2013).





Heating by electric currents, alternatively named frictional and Joule heating, is another important source. The Joule heating rate can be expressed locally as

$$q_{\mathrm{JH}} = \sigma_{\mathrm{P}} \left( \mathbf{E} + \mathbf{u} \times \mathbf{B} \right)^2 \tag{15}$$

with $\sigma_{\mathrm{P}}$ the Pedersen conductivity, $\mathbf{E}$ the electric field, $\mathbf{u}$ the neutral wind speed and $\mathbf{B}$ the magnetic field. The generated heat flux varies from insignificant to about $60\,\mathrm{mW\,m^{-2}}$, the estimated global power between about 1 GW and up to roughly 1 TW (Buchert, in review; Fedrizzi et al., 2012; Sarris et al., 2020), although different methods suggest either large underestimations (Codrescu et al., 1995) or overestimations (Palmroth et al., 2005) within the measurements, while the exact values are not known. The ion–neutral frictional heating flux peaks during large geomagnetic storms (Lu et al., 2016). In such cases, JH

surpasses the thermospheric heating by solar radiation, although on average the solar input is larger. However, JH rates, their spatial distribution and temporal variations are relatively poorly determined, preventing more quantitative assessments.

The third source of heat is associated with particle precipitation which produces the aurora. The total heat flux produced by particle precipitation can range between 50% and 100% of that produced through Joule heating (Vickrey et al., 1982). Large particle precipitation energy fluxes tend to occur on small scales (visible as structures in the aurora); consequently, the energy

flux associated with auroral particle precipitation can often surpass that of Joule heating locally. Virtanen et al. (2018) estimated corresponding values of up to $160\,\mathrm{mW\,m^{-2}}$.

Figure 7 shows the electron density, Pedersen conductivity and Joule heating rate in a TIE-GCM simulation of the LTI on 17 March 2015. The top row shows views above the North Pole while the bottom row shows latitude–altitude cuts (in the direction indicated with red lines in the upper panels). Fig. 7e and Fig. 7f indicate that both the Pedersen conductivity and

the Joule heating rate maximise at about 120 km altitude, whereas Fig. 7b and Fig. 7c indicate that the global maxima of each parameter occurs at different local times. This illustrates the dependence of Joule heating not only on the Pedersen conductivity but also on the electric fields which can be very localised. The white stripes centered on 90° latitude in the bottom-row figures correspond to the region of 87.5–90° latitude where TIE-GCM does not give output.

### 4.8 Heat transfer to the neutral gas by ion and electron cooling

When the ion and electron temperatures, $T_i$ and $T_e$, respectively, are increased compared to the neutral temperature $T_n$, heat is transferred to the neutral gas by ion and electron cooling, and also between ions and electrons. The corresponding steady-state heat transfer rate from ions to neutrals can be expressed as (e.g., Killeen et al., 1984)

$$q_{\Delta T,i} = N_e \nu_{in} \frac{m_i}{m_n + m_i} 3 k_B (T_i - T_n), \tag{16}$$

with $N_e$ the electron density, $\nu_{in}$ the ion–neutral collision frequency, $m_n$ the neutral mass, $m_i$ the ion mass, and $k_B$ Boltz-

mann's constant. Neglecting heat transfer from and to the light electrons, this heat transfer would (for similar ion and neutral compositions) amount to about half of the JH rate (15), while the other half of the JH rate heats the neutrals directly.

The analogous expression for the heat transfer rate to the neutral gas by electron cooling, $q_{\Delta T,e}$, is complicated by the numerous inelastic collisions between electrons and neutrals which are important for energy transfer. Comprehensive tables





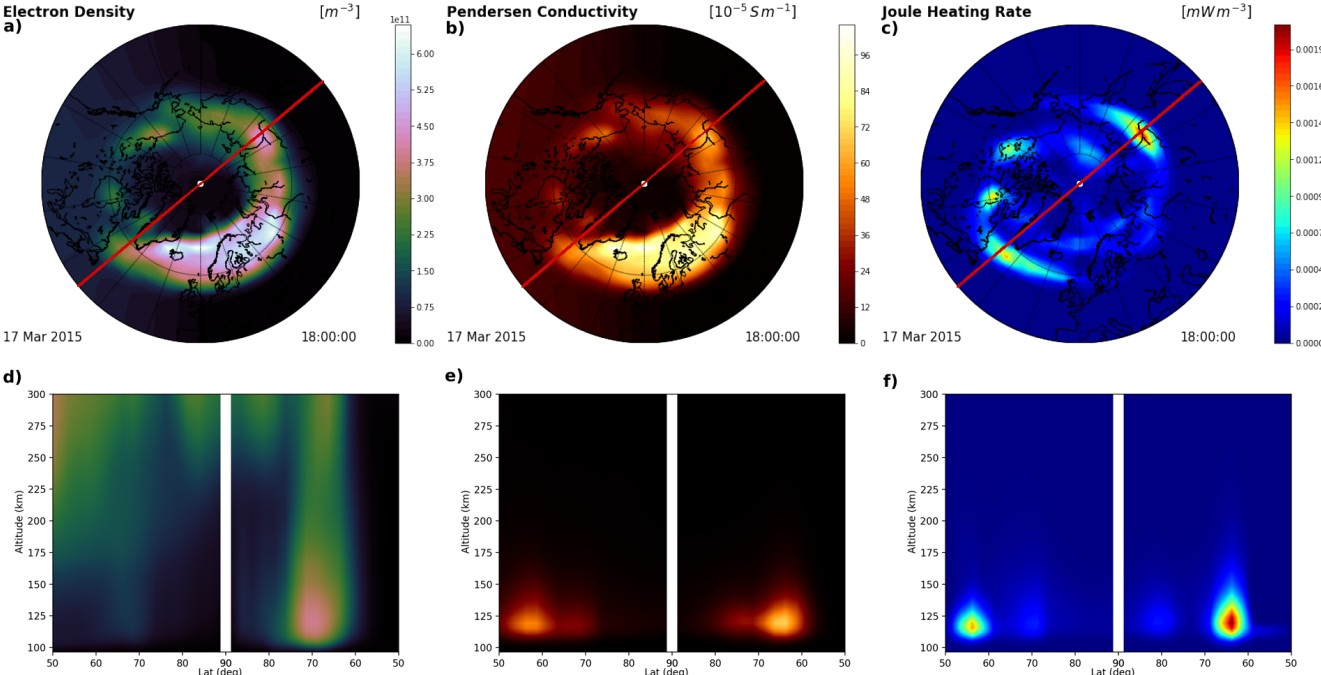

**Figure 7.** Maps of (a) the electron density, (b) the Pedersen conductivity, and (c) the Joule heating rate at ∼120 km altitude on 17 March 2015 18:00:00 (UTC) based on TIE-GCM. (d) Electron density, (e) Pedersen conductivity and (f) Joule heating rate as a function of geographic latitude and altitude in the plane indicated with a red line in the above panels.

of relevant cross sections can be found in Schunk and Nagy (2009). Finally, above roughly 200 km altitude, ion-electron

Coulomb collisions provide for significant heat exchange between ions and electrons when $T_i \neq T_e$. Contrary to the external heating mechanisms discussed in Sect. 4.7, heat transfer between neutrals, ions and electrons is an internal process in the LTI; hence, it is not associated with a heat source from the LTI perspective.

In practice, estimating the amount of heat transferred to the neutral gas requires coordinated measurements and subsequent modelling, and can thus only be carried out in fortuitous conditions at specific locations, where measurements are available.

Such conditions were met for instance in Marchaudon et al. (2018), who used simultaneous measurements from SuperDARN and EISCAT radars, ionosondes, the CHAMP satellite (Reigber et al., 2006), and subsequent modelling with the IPIM model (Marchaudon and Blelly, 2015) to investigate the mechanism behind the ionosphere $F$-peak electron density decrease at high latitudes during a high-speed stream event. They concluded that neutrals in the thermosphere were heated by up to 160 K during the studied event as a consequence of enhanced high-latitude convection, in agreement with earlier results by Gardner

et al. (2012). However, Marchaudon et al. (2018) showed that the long-lasting ionospheric effects were due to the fact that this heating led to the expansion of the thermosphere and a change in the [O]/[N$_2$] ratio at the $F$-region peak altitude, which ultimately resulted in the extinction of the ionospheric $F_2$ layer.





### 4.9 Atmospheric waves

Forcing from below can also affect the derived parameters within the LTI. The most important type of forcing from below
is presented by gravity waves, which are excited by many meteorological processes (Fritts and Alexander, 2003), and which
become increasingly prevalent at higher altitudes due to decreasing air density. The gravity waves, especially the ones with
fast vertical speed, can penetrate into the LTI region and up to ∼200 km before they are dissipated by molecular damping.
These waves perturb the neutral winds and neutral and plasma temperatures and densities. Travelling ionospheric disturbances
(TIDs) have been observed in connection with tornados, deep convections, and earthquakes (Tsugawa et al., 2011; Nishioka
et al., 2013; Azeem et al., 2015, 2018). Gravity waves may seed plasma instabilities, leading to the formation of equatorial
Spread-F (Kelley et al., 1981; Hysell et al., 1990; Palmroth et al., 2000). High-resolution numerical simulations also suggest
that gravity waves can produce large vertical wind shears above the mesopause, where the atmosphere is statistically the
most stable (Liu, 2007, 2017). These wind shears can have important implications for *E*-region electrodynamics, including
sporadic *E* layers (Mathews, 1998; Haldoupis, 2012).

Gravity wave dissipation, either due to wave breaking or molecular damping, causes heat and species transport (Walterscheid,
1981; Liu, 2000; Becker, 2004; Garcia et al., 2007). This affects the transport within the LTI, its mass exchange with the
mesosphere, and the compositional structure of the entire thermosphere. In numerical models, such as WACCM-X and TIE-
GCM, the thermospheric density, the O/$N_2$ ratio, and ionospheric plasma density are sensitively dependent on the effective
eddy diffusion parametrised or specified in the mesosphere and lower thermosphere. For example, rather realistic semi-annual
variation in thermospheric density was obtained by adjusting the eddy diffusion at the lower boundary of TIE-GCM (at ∼97 km)
(Qian et al., 2009).

On global scales, tidal waves and fast-propagating planetary waves (e.g., ultra fast Kelvin waves) and their variability can be
an important cause of LTI variability. Like gravity waves, propagating tides can reach up to ∼200 km before being largely dis-
sipated by molecular damping, and can cause large perturbations in wind, temperature and composition in the LTI. Propagating
tides are quantified by satellites, as outlined in a comprehensive review by Liu (2016). Recent studies have provided further
evidence that the ionospheric day-to-day variability could be closely tied to the tidal day-to-day variability. For example, the
pre-reversal enhancement of equatorial vertical $\mathbf{E} \times \mathbf{B}$ drift shows strong day-to-day variability. A recent analysis demonstrates
that, through the *E*-region dynamo at midlatitudes in the summer hemisphere, the day-to-day variability of tidal winds plays
a central role (Liu, 2020). It is also found that tidal winds determine the day-to-day variability of the equatorial vertical drift
(Zhou et al., 2020).

Contrary to tides, the quantification of gravity waves and of their effects in the LTI remains a challenge for both obser-
vations and numerical modelling, because of the scarcity of LTI observations in general, the multiscale nature of gravity
waves (10s–1000s km), and their global distribution. The altitude range of most ground-based and satellite techniques for
measuring perturbations associated with gravity waves is limited to below 100 km. A notable exception is lidar measure-
ments of metal layers, which can sometimes reach up to ∼170 km (Chu et al., 2016, and references therein). However,
the occurrence of such metal layers is sporadic, and it is challenging to obtain horizontal information, long-term variation,





and global distribution of the waves from these measurements. Two new NASA satellite missions, Global-scale Observations of the Limb and Disk (GOLD; Eastes et al., 2017) and Ionosphere Connection Explorer (ICON; Immel et al., 2018) make remote measurements in the LTI region, around 160 km and 100–160 km, respectively. LTI gravity wave information
can potentially be extracted from their measurements. With the scarcity of direct gravity wave measurements in LTI, low-altitude *in situ* measurements would be extremely valuable. At the same time, global measurements of gravity waves in the mesosphere, for example by the upcoming NASA Atmosphere Wave Explorer (AWE, https://www.nasa.gov/press-release/nasa-selects-mission-to-study-space-weather-from-space-station), will be highly complementary in linking the waves in the LTI to lower atmospheric sources.

**4.10  Total electron content and derived quantities**

The total electron content (TEC) refers to the integrated electron density $n_e$ along the line-of-sight between a receiver and a satellite-borne transmitter: $TEC = \int n_e \, \mathrm{d}s$. It can be derived from the measurements of two carrier wave frequencies $(f_1, f_2)$ transmitted by Global Navigation Satellite System (GNSS) satellites:

$$sTEC = \frac{f_1^2 f_2^2}{f_1^2 - f_2^2} \frac{L_1 - L_2}{K} + DCB_t + DCB_r + \Delta\epsilon, \tag{17}$$

where $L_1$ and $L_2$ are carrier phase observations, $K$ is a constant, $DCB_t$ and $DCB_r$ are differential code biases of the transmitter and receiver, respectively, and $\Delta\epsilon$ accounts for ambiguities due to cycle slip corrections and remaining errors. $sTEC$ is the slant TEC related to the actual number of electrons between the transmitter and receiver. It depends on the elevation angle of the GNSS satellite as the ray path through ionospheric layers with higher electron density gets longer for lower elevations. By using an adequate mapping function, the vertical TEC, $vTEC$, can be obtained (e.g., Noja et al., 2013; Zhang et al., 2016;
Montenbruck and Rodríguez, 2020; Jin et al., 2019). For ground-based receivers, it is often assumed that the pierce point of the ionosphere is at 350 km, and a thin, shell-like ionospheric model is used for obtaining $vTEC$. However, for a receiver on-board another satellite above the pierce point, often a more advanced mapping function, assuming thick ionosphere should be used (Noja et al., 2013). More complex approaches for $vTEC$ are also typically used in tomography techniques (e.g., Meggs et al., 2004). Note that additional biases in TEC measurements, such as receiver temperature, may contribute to errors in TEC
estimations (Coster et al., 2013). The concepts of ground-based $sTEC$ and its mapping to $vTEC$ are shown in Figure 8a.

TEC is nowadays routinely used for monitoring electron density variations in the ionosphere. It is generally displayed as 2D maps based on networks of ground-based receivers (Coster et al., 1992). Over the years, TEC has successfully been used for studies of the response of the ionosphere to geomagnetic storms (Danilov, 2013; Mendillo, 2006), but also to study dynamics of ionospheric structures such as polar cap patches or travelling ionospheric disturbances (Tsugawa et al., 2004; Durgonics et al.,
2017). In numerical simulations which model the LTI, TEC is naturally obtained by integrating the computed electron density along a (slanted or vertical) path. Similarly, from tomographic reconstructions of the ionosphere (e.g., Norberg et al., 2015), TEC can be obtained by numerical integration of the inverted electron density. In such cases, additional measurements than solely from GNSS, such as incoherent scatter radar or ionosonde observations, can be used to feed the tomographic inversion

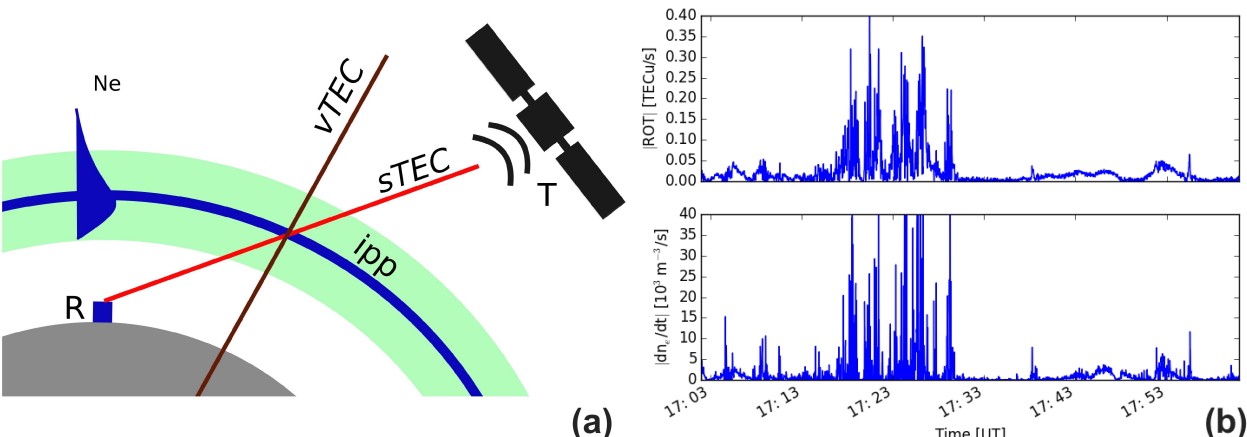

**Figure 8.** (a) Schematics of obtaining $sTEC$ and $vTEC$: T - transmitter, R - receiver, Ne - electron density profile in the ionosphere, ipp - ionospheric pierce point. (b) Example of ROT and electron density variations $dn_e/dt$ (absolute values) as measured by *Swarm A* during one hour on September 8, 2017.

system and therefore contribute to the TEC derivation. One example of such an assimilative tomography network is Tomoscand
in Fennoscandia (e.g., Norberg et al., 2018).

Ground-based networks used for TEC calculations are limited in coverage, in particular in the polar regions and over oceans. This limitation can be addressed by GNSS receivers onboard LEO-satellites. Such satellite-based measurements of the topside ionosphere TEC have recently provided an additional highly valuable contribution, by characterising structures, density gradients and irregularities in ionospheric plasma (e.g., Chartier et al., 2018; Jin et al., 2019). They further support *in situ* plasma
density measurements, as they do not only follow similar characteristics in the context of structuring (Zakharenkova et al., 2016; Xiong et al., 2016; Jin et al., 2019), but they also add additional information about the directional extent of plasma density variations (e.g., Park et al., 2017b; Follestad et al., 2020). Thus TEC, together with *in situ* plasma density measurements, gives a detailed insight into scales of plasma density structures, gradients, and the overall plasma density variations. A large number of tracked GNSS satellites by a receiver onboard LEO-satellites can even provide insight into the spatial extent
of structures in the ionosphere, such as polar cap patches, equatorial Spread-F, auroral blobs, when using inversion methods (Follestad et al., 2020).

Derived TEC parameters, such as the rate of change of TEC (ROT), being the temporal derivative of TEC, and the ROT index (ROTI), being the standard deviation of ROT in a given time interval, provide important characterisation of plasma irregularities in the topside ionosphere. It has been shown that satellite-based TEC variations are correlated with the *in situ*
measurements of plasma density variations (Xiong et al., 2016). Indeed, due to the shape of ionospheric density profiles, the largest contribution to TEC comes from plasma in the vicinity of the LEO satellite. Thus, ROT and ROTI are closely related to the variability in the local ionospheric plasma density (Jin et al., 2019; Xiong et al., 2016), see also Fig. 8b. As such, they can also be related to the quality of trans-ionospheric radio signals, and since TEC is derived from the measurements of GNSS





signals, these measurements can also give insight into processes behind ionospheric scintillations, especially if the receiver
provides high-frequency data.

TEC and other GNSS-related measurements depend on the electron density variations in the topside ionosphere, and hence
on the satellite's orbit. The peak of the ionospheric $F$ layer, which gives the largest contribution to the variations in the signal
propagation, is below the altitudes of most of LEO-satellites. A low-perigee satellite mission embedding a high-frequency
GNSS receiver onboard could allow for the first time the probing of different contributions to TEC, in particular from different
parts of the $F$ region and upper $E$ region of the ionosphere or from the plasmasphere, and related variability, and as such it
could provide a unique opportunity to investigate the effect of plasma irregularities on the GNSS signals at different altitudes.

## 5   Concluding remarks

There are at least four viewpoints from which understanding of the LTI is crucial: scientific, policy-based, technological, and
from the perspective of the society at large. First and foremost, humans explore – not only our own planet but also celestial and
astronomical bodies within and beyond our Solar system. It is astonishing that we can find a region so near to the surface of our
planet – 80 to 200 km altitude – which is still called the ignorosphere due to the lack of systematic *in situ* measurements. Since
the LTI is shown to influence atmospheric and climate systems, it should be understood as part of our planetary system. In fact,
from this perspective the entire near-Earth space linking to the LTI belongs to the Earth system. Even so, the LTI and the near-
Earth space are often regarded as fall-betweens as the understanding of their phenomena requires knowledge from atmospheric
physics, chemistry and space physics. This multitude of fields should not hinder us in exploring the LTI; on the contrary the
inherent interdisciplinary nature should challenge us to think about the region from many different scientific perspectives.

Policymakers are continually interested in the climate system due to climate change and its anthropogenic nature. From the
climate change perspective, as described in Sect. 2.3, it is imperative to understand the role of electron precipitation within the
natural polar climate variability to accurately quantify the contribution due to human activities. Understanding the LTI is a key
element in this endeavour. However, policymakers have recently also been pushing towards getting prepared for possible large
space weather events, as, e.g., the European Union, the United Kingdom and the United States of America have issued reports
in this matter (see, e.g., Cabinet Office, 2017; Executive Office of the President of the United States, 2019). The goal is to
develop mitigation strategies to protect human infrastructure from the potentially devastating impact that a major geomagnetic
storm could have. The current understanding of geospace still contains some gaps that preclude predictive simulations of the
effects to be anticipated from a given solar storm. One such gap is related to the LTI, whose complexity comes from the
intertwined behaviours of its neutral and ionised components which are governed by distinct physical processes and from its
couplings to below and above. A major difficulty lies in that it proves extremely challenging to measure physical parameters at
LTI altitudes, both through *in situ* instruments and via remote observations.

From a technological perspective, the reason why the LTI is called the ignorosphere is because the exploration of the region
is so difficult. Spacecraft on circular low-Earth orbits experience thermal problems and return to Earth due to the increased
drag. Remote sensing methods require some emission to be gathered by the remote instrument, but there are regions within





the LTI which do not emit these signals (Sarris, 2019). Ground-based measurements, such as the incoherent scatter radars are invaluable in describing the ionised part of the LTI. However, due to the coupled nature of the LTI, it is necessary to explore several other regions such as the magnetotail simultaneously, which is of course extremely challenging to achieve in a systematic way. Therefore the exploration of the LTI needs to be understood as a coordinated effort from the beginning. However, due to the difficulties involved, a mission obtaining systematic *in situ* measurements of the LTI would be a great technological achievement comparable to exploring the deepest seas and the furthest galaxies.

In this review, open questions related to the LTI have been divided according to three viewpoints: energetics, variability and dynamics, and chemistry. From the perspective of energetics, large uncertainties exist in estimating the relative contributions of the various sources of energy (solar radiation flux, Joule heating, particle precipitation, atmospheric waves, exothermic chemical reactions) under various contexts (latitude, local time, season, geomagnetic activity, solar cycle), which makes the determination of the LTI energy budget very challenging. Regarding variability and dynamics, the understanding of the complex couplings of the LTI with the magnetosphere and the underlying neutral atmosphere is still limited. This is not only true at high latitudes, where substorms create large effects, but also at low latitudes where the equatorial electrojet and plasma irregularities play a large role. Finally, the LTI chemistry involves a great number of species, whose densities are determined by temperature-dependent photochemical reactions, and by production rates associated with particle precipitation. However, many of the critical parameters, such as the ion–neutral cross sections and collision rates, remain poorly characterised.

To address the unanswered science questions pertaining to the LTI, additional observations of its key parameters, ideally *in situ*, are needed in various regions of interest. In the ionospheric $E$ region, interest is in the equatorial electrojet region, corresponding to geomagnetic latitudes (MLAT) comprised within, roughly, $\pm 7°$, and the auroral latitudes, comprised in most situations within $60$–$75°$. In the $F$ region, the areas of interest encompass all latitudes and include, in particular: equatorial plasma bubble region (within $\pm 30°$ MLAT, 18–4 MLT), midlatitude $S_q$ currents (within $\pm 60°$ MLAT, 6–19 MLT), auroral latitudes, the polar cusp regions on the dayside (70–80° MLAT, 10–14 MLT), and the polar cap region ($> 70°$ MLAT). Ideally, systematic observations in these regions should cover a wide range of seasons and geomagnetic conditions to enable the study of their effects on the LTI energetics, variability, dynamics, and chemistry.

Systematic *in situ* observations of the key LTI parameters (precipitating particle fluxes and energy spectra, ion and neutral temperatures, compositions and densities, neutral winds, ion drift speeds and electric fields, and magnetic fields) could allow the derivation of physical quantities whose knowledge is crucial to the correct modelling of the LTI and its external drivers. For instance, the forcing from above consists not only of particle precipitation but also of the Lorentz force exerted onto the ions, of field-aligned currents, and of the Poynting flux which all need to be taken into account while developing boundary conditions for ionospheric models. Microscopic parameters such as the ion–neutral cross sections, collision frequencies, and heat transfer coefficients are also needed in the kinetic transport and photochemical modules of ionospheric models. Furthermore, supercomputing centres are becoming increasingly accessible to research groups, and hence space environment modelling is expected to make significant steps towards understanding and forecasting space weather. The increased amount of available computing power also enables assimilative schemes to incorporate more data, as is currently done using the AMIE





technique for ground-based radar and satellite observations (Cousins et al., 2015). All these models benefit from systematic *in situ* observations, leading to a reduction of uncertainties and errors in numerical simulations.

Finally, a phenomenon that continually interests society at large is the beautiful and vivid auroral displays that take place in the LTI. For example, a recent Citizen Science auroral discovery, named *the dunes*, reached 2.5 billion people all around

the world in just two weeks after publication (Palmroth et al., 2020). This discovery followed a similar one by a collaboration between Canadian amateur photographers and space physicists, which revealed the phenomenon nowadays known as STEVE (MacDonald et al., 2018; Gallardo-Lacourt et al., 2018), and which has reached even larger numbers describing public interest. All above-mentioned features pertaining to the LTI – its role as a region to be explored, the technological challenges it poses, and its role within world-wide policies combined with the fascinating aurora and other optical features – makes it an ideal topic

for a wide range of stories to engage the public at large, generating increased interest in natural sciences. For these reasons, a dedicated satellite mission providing regular observations of the key LTI parameters is highly desirable and timely.

*Code and data availability.* The DMSP data used to make Figure 3a are provided by NOAA and can be downloaded from the DMSP/SSJ archives at https://satdat.ngdc.noaa.gov/dmsp/data/. Figure 3b was made using OvationPyme, the Python implementation of Ovation Prime 2010, which is available at https://github.com/lkilcommons/OvationPyme. The NRLMSISE-00 and IRI-2012 models used to produce Fig-

ure 4 are available online through NASA's Community Coordinated Modeling Center (CCMC) at https://ccmc.gsfc.nasa.gov/modelweb/ models/nrlmsise00.php and https://ccmc.gsfc.nasa.gov/modelweb/models/iri2012_vitmo.php, respectively. Data from the Swarm mission visualised in Figures 5 and 8(b) are provided by ESA via http://swarm-diss.eo.esa.int or ftp://swarm-diss.eo.esa.int. TIE-GCM model run data used to generate Figure 7 were produced at the Democritus University of Thrace, with AMIE inputs provided by Gang Lu, NCAR/HAO. WACCM-X model run data used to generate Figures 1 and 2 were produced at NCAR/HAO, provided by Federico Gasperini. Both data sets

are available from the author TS on request, while run requests of TIE-GCM can be made via the CCMC at https://ccmc.gsfc.nasa.gov/ models/modelinfo.php?model=TIE-GCM and the WACCM-X code can be downloaded as part of the Community Earth System Model version 2 http://www.cesm.ucar.edu/models/cesm2/.

*Author contributions.* The writing of this article was led by MP and coordinated by MG, with contributions from all co-authors. ED made Figures 1, 2, and 6. MG made Figure 3, RP made Figure 4, CSt and NO made Figure 5, TS and ST made Figure 7, WM made Figure 8.

Significant contributions to the text by section are as follows: 1: MP; 2.1: MP, SB, HLL; 2.2: MP, DM, CSt, MY; 2.3: MP, PTV, MAC, MG; 3.1: MG, AK; 3.2: PV, GM, MG; 3.3: ID, PV, MG; 3.4: GM, GK, MG, ED; 3.5: AM, AA, DK, MP; 3.6: CSt, NO; 4.1: OM, MP, TMJ, DK; 4.2: DK, TMJ, OM, MP, MG; 4.3: MP, DK; 4.4: MG, MY; 4.5: MG, DK; 4.6: TM, MG, AA; 4.7: SB, MG; 4.8: SB, MG; 4.9: HLL, MP; 4.10: WM, CSi, JvdIJ, MG; 5: MP, MG.

*Competing interests.* The authors declare that they have no competing interests.





*Acknowledgements.*  This work was funded under the European Space Agency (ESA/ESTEC) contract number 4000127346/19/NL/IA with
the Democritus University of Thrace for the Daedalus science and requirements consolidation study, in the framework of the Earth Ex-
plorer 10 Phase-0 feasibility studies (PI: TS). MP and MG are supported by European Research Council Consolidator grant 682068-
PRESTISSIMO, and Academy of Finland grants 309937 and 312351. Work at the Democritus University of Thrace is supported under
project KE82324. National Center for Atmospheric Research is a major facility sponsored by the National Science Foundation under Co-
operative Agreement No. 1852977. HLL's work is in part supported by NSF grants OPP-1443726 and AGS-1552153 and NASA grant
NNX16AB82G. AM thanks the Programme National Soleil-Terre de l'Institut des Sciences de l'Univers (PNST/INSU) for the scientific
support of her activities. WJM is supported in part by the Research Council of Norway grants 267408 and 275653. OM acknowledges sup-
port by ESA contracts 4000127660 MAGICS and 4000118383 SIFACIT. The WACCM-X model output visualised in Figures 1, 2 and 9 was
provided by Federico Gasperini (NCAR/HAO) and the model simulations made use of AMIE high-latitude forcing input provided by Gang
Lu (NCAR/HAO).





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
