# Peer review of "Lower thermosphere - ionosphere (LTI) quantities: Current status of measuring techniques and models"

_Annales Geophysicae, 2020_

## Referee Comment (RC1) · Anonymous Referee #1 · 9 Aug 2020

Overall quality of the manuscript (general comments):

The manuscript "Lower thermosphere - ionosphere (LTI) quantities: Current status of measuring techniques and models" by Plamroth et al. is a well written review of the lower thermosphere and ionosphere, with the focus on the open questions with a lot of them cannot be addressed without high-quality in-situ measurements and describing the state of the art of measuring this region. The manuscript is motivated by the Daedalus mission. In my opinion the manuscript is a very valuable contribution to the literature of the LTI region and will most likely be used by many researchers.

Addressing individual scientific questions/issues ("specific comments"):

[Figure]

line 5: should the wind dynamo be mentioned?

Line 8: does directly mean "in-situ" or without any elaborate assumptions?

Line 17: I think the mesosphere was originally termed "ignorosphere" since it is too low for satellites and too high for airplanes and weather balloons. So maybe change "this region" to the "LTI region"

Line 44: "the motion of the atmosphere is driven by both solar irradiance and waves." Do the authors mean thermal atmospheric tides caused by solar irradiance? Maybe reformulated so that it fits to the waves.

Line 85: There are other efforts of whole atmosphere models: WAM (), GAIA ().

Figure 2: Is the depicted neutral wind the total horizontal wind? Similar for the ion drift- is this the ExB drift and is the one perpendicular to the magnetic field?

Line 199: "total upward energy flux by resolved waves at 100 km" Does this refer to only 100km waves or also larger ones?

Line 201: "horizontal scales less than 200 km are poorly resolved" Shouldn't waves be resolved with wavelengths approximately 4x the resolution? How does this fit to the 100km in line 199?

Line 518: "resolution ranges from one orbit to several days" It is not clear to me what is meant here? Orbit averaged to several day averaged? Is this temporal resolution-one measurement pre orbit or every few days?

Line 661: Maybe the Weimer (2005) empirical ion convection model based on DE-2 data could be mentioned

Line 715: section 3.6 Magnetic fields: I may have missed it but the summary does not mention that the Swarm was able to derive currents without any assumption of current flow due the constellation with nearby satellites. I find this an important point since at the end of the section, the E-region is mentioned and this is the region where

strong currents flow. So the interpretation of magnetic fields with respect to current flow without constellation is a challenge (e.g. see the modeling of Maute & Richmond 2017). Maute, A., Richmond, A.D. F-Region Dynamo Simulations at Low and Mid-Latitude. Space Sci. Rev 206, 471–493 (2017).

Line 757: "which are essential also for FACs" It is not quite clear to me what this means? That FAC flows along magnetic fieldlines?

Line 773: "Above the E-layer, electrons and ions drift together and the ionospheric current vanishes." I do not think the authors mean that there is no ionospheric current above the E-region as the sentence suggests. Could this be further explained?

Technical corrections at the very end ("technical corrections")

Line 125: Fig 2. Remind the usual -> reminds of the

Figure 1: I suggest to add the altitude or pressure range of the plots. Does it go from the surface to approximately 500km?

Figure 2: It would be easier to add approximate solar local times to the geographic locations of the profiles in the captions.

Line 146: reference frames with the neutral gas velocity -> of the neutral gas?

Line 212: "In this topic, the" Should this read "in this study/review"?

Line 300: suggested: to the lower atmosphere

564: Should planetary waves be mentioned?

Line 600: Any reason to use speed instead of velocity?

Line 648: metre-> meter

Line 449: analyser -> analyzer (at least that is the spelling UTD is using)

Line 835: engineering grade magnetometer

Line 1083: "propagating tides" upward propagating? Tides can also propagate eastward and westward. Not all tides reach the F-region.

---

## Referee Comment (RC2) · Anonymous Referee #2 · 28 Sep 2020

The article "Lower thermosphere - ionosphere (LTI) quantities: Current status of measuring techniques and models" by M. Palmroth et al. constitutes a large review of the complex physical environment of Earth Lower thermosphere and ionosphere, covering theoretical considerations, experimental techniques and models. The term ignorosphere has been used to indicate this part of the Earth outer space, because of the difficulties of establishing measurement systems that could monitor its physical and chemical parameters on a continuous base globally. Therefore there are still many open scientific questions, that the proposed Daedalus mission could help answering from in-situ measurements. In the Lower thermosphere the forcings from above (principally solar radiation and particle precipitations) deposit large amounts of energy and

the forcings from below (principally through tides and waves activity) can significantly affect the physico-chemical processes of both thermosphere and ionosphere. There are such a large varieties of phenomena taking place in this region, that it is not possible to resume all of them in a single review paper, and this paper covers most of them, also pointing out to some phenomena of interest also outside the scientific community, with recently discovered auroral activities. In the article various measuring techniques are described, covering not only thew Lower Thermosphere region, but the thermosphere and ionosphere as a whole, with a specific focus to point out the current limitations of this lower altitudinal range and expressing the need for new satellite missions devoted to collect in-situ measurements of the various quantities involved: particles, densities, winds, magnetic fields... Despite the complexity of the topics covered, the article is well organised and refers to recent advances in all the disciplines involved, both from the theoretical and experimental point of view.

The article can be accepted for publication.

The following remarks are suggestions for minor corrections that could be implemented for improving the overall quality of the manuscript.

Reading the manuscript it clearly appears that it has been written by multiple authors: a few definitions are expressed a couple of times, but without significant overlap.

Acronyms are not always defined at first use, sometimes they are defined multiple times e.g. (list non exhaustive): WACCM on page 4, 8, 16 and 19 for WACCM-X, which was already defined on page 4 GPS used on page 19, defined on page 20. GNSS defined on page 39, used on page 19, 20. IPIM defined on page 16 and 19 SuperDARN defined on page 31, used first time on page 23. TomoScand is not defined on page 40.

Some figures show ionospheric parameters computed using the International Reference Ionosphere model (IRI). The citation provided and the model shown are not the latest version, which was released in 2016, IRI-2016. The profiles shown might be identical in the IRI-2016, but the latest publications of this model should be cited

(e.g. on page 4 and 17): Bilitza, D., D. Altadill, V. Truhlik, V. Shubin, I. Galkin, B. Reinisch, and X. Huang (2017), International Reference Ionosphere 2016: from ionospheric climate to real-time weather predictions, Space Weather, 15, 418–429, doi:10.1002/2016SW001593.

I think that captions of figures 1 and 2 should be improved. it is not clear if these two figures show the same altitude range: figure 1 is representing an isobaric surface, while figure 2 a fixed geometrical altitude range. In the text it is indicate that WACCM-X has been extended up to about 500 km, but an explicit indication in the figure caption would be helpful. In the representation of the quantities shown in figure 1, there are no axes to allow the reader to understand where the various panel's quantities are located in altitude.

On page 11 line 286, the need of a sufficient horizontal resolution and time resolution at low latitudes is expressed, but it is not quantified. I suggest to provide some values that could be used as guidelines for reaching specific scientific goals. This same remark is valid for other conclusions of this article, where no explicit values are indicated.

Figure 4 shows combined results from two different empirical climatological models: NRLMSISE-00 for the neutrals and IRI for the ions. it shows that for some species (N and O) the scale heights of the neutral and its corresponding ion are extremely different. It turns out that in the upper part of the plot the density differences can be of many orders of magnitude. This points out clearly the limitations of these models and an indication that in-situ measurements of both neutrals and ions are necessary.

Page 38, line 1069: following earthquakes, even tsunamis are source of gravity waves observed in the ionosphere (e.g. Makela, J. J., P. Lognonné, H. Hébert, T. Gehrels, L. Rolland, S. Allgeyer, A. Kherani, G. Occhipinti, E. Astafyeva, P. Coïsson, A. Loevenbruck, E. Clévédé, M. C. Kelley, and J. Lamouroux (2011), Imaging and modeling the ionospheric airglow response over Hawaii to the tsunami generated by the Tohoku earthquake of 11 March 2011, Geophys. Res. Lett., 38(13), L00G02). On line 1065

the focus of this part has been put on gravity waves, but some of the phenomena high-
lighted produce mostly acoustic waves. It is stated correctly that TIDs are observed,
which encompass both kind of waves. I think that the unaware reader might not per-
ceive their differences.

---

## Author Comment (AC1) · 26 Nov 2020

Dear Reviewer #1,

Thank you for your thorough review of our paper and for your positive and constructive comments. We have addressed all of your very good points which we think will significantly improve the quality of the paper. We go through the questions in the below separate document, where your original comments are given in boldface font.

On behalf of the co-authors, Minna Palmroth

[Figure]

Please also note the supplement to this comment:
https://angeo.copernicus.org/preprints/angeo-2020-42/angeo-2020-42-AC1-
supplement.pdf

**Supplement:**

**Manuscript angeo-2020-42 "Lower thermosphere – ionosphere (LTI) quantities: Current status of measuring techniques and models" by Palmroth et al.**

**Response to Reviewer #1**

We thank the Reviewer for their very positive and constructive comments on our manuscript. Below we present how we intend to address them in a revision of the paper. The Reviewer's comments are reproduced in bold font, and our responses are given in normal font.

**Overall quality of the manuscript (general comments):**

**The manuscript "Lower thermosphere - ionosphere (LTI) quantities: Current status of measuring techniques and models" by Palmroth et al. is a well written review of the lower thermosphere and ionosphere, with the focus on the open questions with a lot of them cannot be addressed without high-quality in-situ measurements and describing the state of the art of measuring this region. The manuscript is motivated by the Daedalus mission. In my opinion the manuscript is a very valuable contribution to the literature of the LTI region and will most likely be used by many researchers.**

Thank you very much for this very positive opening statement.

**Addressing individual scientific questions/issues ("specific comments"):**

**Line 5: should the wind dynamo be mentioned?**

This is a very good point, we will add a few sentences on the wind dynamo in the revision.

**Line 8: does directly mean "in-situ" or without any elaborate assumptions?**

Indeed, here we the intended meaning is "in situ". We can replace "directly" by a more accurate phrase in the revised version of the manuscript to avoid any ambiguity.

**Line 17: I think the mesosphere was originally termed "ignorosphere" since it is too low for satellites and too high for airplanes and weather balloons. So maybe change "this region" to the "LTI region"**

The Reviewer is right, we will correct this in the revision and make sure we avoid using "ignorosphere" when referring to the LTI.

**Line 44: "the motion of the atmosphere is driven by both solar irradiance and waves." Do the authors mean thermal atmospheric tides caused by solar irradiance? Maybe reformulated so that it fits to the waves.**

Indeed, thank you for this notion; both are of course driven by solar irradiance in the end. We will reformulate this sentence in the revision.

**Line 85: There are other efforts of whole atmosphere models: WAM (), GAIA ().**

Thank you for pointing to these models, they will be briefly introduced in the revision.

**Figure 2: Is the depicted neutral wind the total horizontal wind? Similar for the ion drift-is this the ExB drift and is the one perpendicular to the magnetic field?**

The neutral winds depicted in this figure correspond to those also shown in Fig. 1, i.e., neutral wind along the model pressure levels. Since the pressure levels are 3D surfaces, there is a small vertical pressure gradient, hence inducing a small component of vertical wind. However, overall the shown neutral wind magnitudes essentially represent horizontal winds.

Regarding ion drifts, they are indeed given by ExB. We are currently working on updating Figs 1 and 2 for the revised manuscript, to address the technical comments made by both Reviewers. Clarifications will be added to the figure descriptions in the revision.

**Line 199: "total upward energy flux by resolved waves at 100 km" Does this refer to only 100km waves or also larger ones?**

Here, 100 km refers to the altitude where the waves are considered, not their wavelengths. We will rephrase this sentence to remove the ambiguity and add a statement regarding the resolved wavelengths.

**Line 201: "horizontal scales less than 200 km are poorly resolved" Shouldn't waves be resolved with wavelengths approximately 4x the resolution? How does this fit to the 100km in line 199?**

Once the ambiguity in the previous sentence is removed, this statement will no longer seemingly conflict with it.

**Line 518: "resolution ranges from one orbit to several days" It is not clear to me what is meant here? Orbit averaged to several day averaged? Is this temporal resolution-one measurement pre orbit or every few days?**

Thank you for this comment, which calls for a clarification. This technique intrinsically allows to infer changes in the neutral density based on the orbit tracking of a single object with a temporal resolution of the order of three days or longer (Doornbos et al., 2008). However, by combining orbit data from multiple tracked objects, it is possible to obtain information on neutral density changes at a resolution of 3 h (Storz et al., 2005). We will modify the original statement in the revised version of the manuscript.

**Line 661: Maybe the Weimer (2005) empirical ion convection model based on DE-2 data could be mentioned.**

Thank you for suggesting this addition, which we will gladly include in the revised manuscript.

**Line 715: section 3.6 Magnetic fields: I may have missed it but the summary does not mention that the Swarm was able to derive currents without any assumption of current flow due the constellation with nearby satellites. I find this an important point since at the end of the section, the E-region is mentioned and this is the region where strong currents flow. So the interpretation of magnetic fields with respect to current flow without constellation is a challenge (e.g. see the modeling of Maute & Richmond 2017). Maute, A., Richmond, A.D. F-Region Dynamo Simulations at Low and Mid-Latitude. Space Sci. Rev 206, 471–493 (2017).**

Thank you for the suggestion, we will make sure to include and discuss this point in the revision.

**Line 757: "which are essential also for FACs" It is not quite clear to me what this means? That FAC flows along magnetic field lines?**

Indeed, the sentence needs rephrasing. We will find a better formulation in the revision (for instance "along which FACs flow") to improve clarity.

**Line 773: "Above the E-layer, electrons and ions drift together and the ionospheric current vanishes." I do not think the authors mean that there is no ionospheric current above the E-region as the sentence suggests. Could this be further explained?**

This is an excellent point; the above statement is misleading and might suggest that F-region currents do not exist. We will correct this by reformulating in the revision.

**Technical corrections at the very end ("technical corrections")**

**Line 125: Fig 2. Remind the usual -> reminds of the**

Will be corrected.

**Figure 1: I suggest to add the altitude or pressure range of the plots. Does it go from the surface to approximately 500km?**

Great suggestion, we will update Fig. 1 and find a way to add altitude/pressure range information.

**Figure 2: It would be easier to add approximate solar local times to the geographic locations of the profiles in the captions.**

Thank you for this suggestion, we will add this information as well.

**Line 146: reference frames with the neutral gas velocity -> of the neutral gas?**

We meant here "other reference frames with non-zero neutral gas velocity U"; we will therefore clarify by rephrasing this in the revision.

**Line 212: "In this topic, the" Should this read "in this study/review"?**

Actually, we should here rephrase into "The current key research question associated to this topic..." to convey our intended meaning.

**Line 300: suggested: to the lower atmosphere**

Indeed, we will add "the" in the revision.

**Line 564: Should planetary waves be mentioned?**

Yes, we will add them in the revised manuscript. Thank you for the suggestion.

**Line 600: Any reason to use speed instead of velocity?**

Thank you for this comment; we should actually rather use "velocity" throughout this section. This will be changed in the revision.

**Line 648: metre-> meter**

**Line 449: analyser -> analyzer (at least that is the spelling UTD is using)**

Indeed, although we are using British English throughout the manuscript, we agree that for instrument names we should follow their preferred spelling. These will be corrected in the revision.

**Line 835: engineering grade magnetometer**

We will implement as suggested in the revision.

**Line 1083: "propagating tides" upward propagating? Tides can also propagate east-ward and westward. Not all tides reach the F-region.**

Thank you for pointing this out; we will modify as suggested to avoid ambiguity.

On behalf of the co-authors,

Minna Palmroth

---

## Author Comment (AC2) · 26 Nov 2020

Dear Reviewer #2,

Thank you for your thorough review of our paper and for your positive and constructive comments. We have addressed all of your very good points which we think will significantly improve the quality of the paper. We go through the questions in the below separate document, where your original comments are given in boldface font.

On behalf of the co-authors, Minna Palmroth

[Figure]

Please also note the supplement to this comment:
https://angeo.copernicus.org/preprints/angeo-2020-42/angeo-2020-42-AC2-supplement.pdf

─────────────────────────

**Supplement:**

**Response to Reviewer #2**

We thank the Reviewer for their very positive and constructive comments on our manuscript. Below we present how we intend to address them in a revision of the paper. The Reviewer's comments are reproduced in bold font, and our responses are given in normal font.

**The article "Lower thermosphere - ionosphere (LTI) quantities: Current status of measuring techniques and models" by M. Palmroth et al. constitutes a large review of the complex physical environment of Earth Lower thermosphere and ionosphere, covering theoretical considerations, experimental techniques and models. The term ignorosphere has been used to indicate this part of the Earth outer space, because of the difficulties of establishing measurement systems that could monitor its physical and chemical parameters on a continuous base globally. Therefore there are still many open scientific questions, that the proposed Daedalus mission could help answering from in-situ measurements. In the Lower thermosphere the forcings from above (principally solar radiation and particle precipitations) deposit large amounts of energy and the forcings from below (principally through tides and waves activity) can significantly affect the physico-chemical processes of both thermosphere and ionosphere. There are such a large varieties of phenomena taking place in this region, that it is not possible to resume all of them in a single review paper, and this paper covers most of them, also pointing out to some phenomena of interest also outside the scientific community, with recently discovered auroral activities. In the article various measuring techniques are described, covering not only the Lower Thermosphere region, but the thermosphere and ionosphere as a whole, with a specific focus to point out the current limitations of this lower altitudinal range and expressing the need for new satellite missions devoted to collect in-situ measurements of the various quantities involved: particles, densities, winds, magnetic fields... Despite the complexity of the topics covered, the article is well organised and refers to recent advances in all the disciplines involved, both from the theoretical and experimental point of view.**

**The article can be accepted for publication.**

**The following remarks are suggestions for minor corrections that could be implemented for improving the overall quality of the manuscript.**

Thank you very much for those encouraging words and for the suggestions!

**Reading the manuscript it clearly appears that it has been written by multiple authors: a few definitions are expressed a couple of times, but without significant overlap.**

Thank you for pointing this out, we will try to harmonise the style and avoid redundancy in the revision.

**Acronyms are not always defined at first use, sometimes they are defined multiple times e.g. (list non exhaustive): WACCM on page 4, 8, 16 and 19 for WACCM-X, which was already defined on page 4 GPS used on page 19, defined on page 20. GNSS defined on page 39, used on page 19, 20. IPIM defined on page 16 and 19 SuperDARN defined on page 31, used first time on page 23. TomoScand is not defined on page 40.**

Thank you for noting these, we will pay close attention to acronym usage and definitions when preparing the revision.

**Some figures show ionospheric parameters computed using the International Reference Ionosphere model (IRI). The citation provided and the model shown are not the latest version, which was released in 2016, IRI-2016. The profiles shown might be identical in the IRI-2016, but the latest publications of this model should be cited (e.g. on page 4 and 17): Bilitza, D., D. Altadill, V. Truhlik, V. Shubin, I. Galkin, B. Reinisch, and X. Huang (2017), International Reference Ionosphere 2016: from ionospheric climate to real-time weather predictions, Space Weather, 15, 418–429, doi:10.1002/2016SW001593.**

Thank you for raising this point; we will replace the reference with the latest one.

**I think that captions of figures 1 and 2 should be improved. it is not clear if these two figures show the same altitude range: figure 1 is representing an isobaric surface, while figure 2 a fixed geometrical altitude range. In the text it is indicate that WACCM-X has been extended up to about 500 km, but an explicit indication in the figure caption would be helpful. In the representation of the quantities shown in figure 1, there are no axes to allow the reader to understand where the various panel's quantities are located in altitude.**

We will indeed update Figures 1 and 2 in the revised manuscript, following suggestions for improvement from both Reviewers.

**On page 11 line 286, the need of a sufficient horizontal resolution and time resolution at low latitudes is expressed, but it is not quantified. I suggest to provide some values that could be used as guidelines for reaching specific scientific goals. This same remark is valid for other conclusions of this article, where no explicit values are indicated.**

This is an excellent point; indeed it would be very valuable to be able to refer to this paper for requirements in the future. We will take this into account in the revision and provide quantified needs wherever possible.

**Figure 4 shows combined results from two different empirical climatological models: NRLMSISE-00 for the neutrals and IRI for the ions. It shows that for some species (N and O) the scale heights of the neutral and its corresponding ion are extremely different. It turns out that in the upper part of the plot the density differences can be of many orders of magnitude. This points out clearly the limitations of these models and an indication that in-situ measurements of both neutrals and ions are necessary.**

Thank you for this great comment, we will emphasise this aspect in the revision.

**Page 38, line 1069: following earthquakes, even tsunamis are source of gravity waves observed in the ionosphere (e.g. Makela, J. J., P. Lognonné, H. Hébert, T. Gehrels, L. Rolland, S. Allgeyer, A. Kherani, G. Occhipinti, E. Astafyeva, P. Coïsson, A. Loevenbruck, E. Clévédé, M. C. Kelley, and J. Lamouroux (2011), Imaging and modeling the ionospheric airglow response over Hawaii to the tsunami generated by the Tohoku earthquake of 11 March 2011, Geophys. Res. Lett., 38(13), L00G02).**

Thank you for this suggestion, we will add tsunamis as additional sources of gravity waves (including the provided reference).

**On line 1065 the focus of this part has been put on gravity waves, but some of the phenomena highlighted produce mostly acoustic waves. It is stated correctly that TIDs are observed, which**

**encompass both kind of waves. I think that the unaware reader might not perceive their differences.**

Thank you for pointing this; we will reword this paragraph to distinguish acoustic waves and gravity waves and be more accurate in their discussion. Further, we will emphasise that the link between the TIDs and gravity waves is one of the objectives for Daedalus.

On behalf of the co-authors,

Minna Palmroth

---

## Author Response (AR1)

**Manuscript angeo-2020-42 "Lower thermosphere – ionosphere (LTI) quantities: Current status of measuring techniques and models" by Palmroth et al.**

**Response to Reviewer #1**

We thank the Reviewer for their very positive and constructive comments on our manuscript. Below we present how we implemented them in the revision of the paper. The Reviewer's comments are reproduced in bold font, and our responses are given in normal font.

**Overall quality of the manuscript (general comments):**

**The manuscript "Lower thermosphere - ionosphere (LTI) quantities: Current status of measuring techniques and models" by Palmroth et al. is a well written review of the lower thermosphere and ionosphere, with the focus on the open questions with a lot of them cannot be addressed without high-quality in-situ measurements and describing the state of the art of measuring this region. The manuscript is motivated by the Daedalus mission. In my opinion the manuscript is a very valuable contribution to the literature of the LTI region and will most likely be used by many researchers.**

Thank you very much for this very positive opening statement.

**Addressing individual scientific questions/issues ("specific comments"):**

**Line 5: should the wind dynamo be mentioned?**

This is a very good point, we added a mention to the wind dynamo.

**Line 8: does directly mean "in-situ" or without any elaborate assumptions?**

Indeed, here the intended meaning is "in situ", hence we replaced accordingly.

**Line 17: I think the mesosphere was originally termed "ignorosphere" since it is too low for satellites and too high for airplanes and weather balloons. So maybe change "this region" to the "LTI region".**

The Reviewer is right: the concept of "ignorosphere" often refers to altitudes encompassing the mesosphere and part of the lower thermosphere. We have rephrased the statement originally on l. 17 and followed the suggestion. We have also slightly adapted the occurrences in the "Concluding remarks" section where "ignorosphere" was used.

**Line 44: "the motion of the atmosphere is driven by both solar irradiance and waves." Do the authors mean thermal atmospheric tides caused by solar irradiance? Maybe reformulated so that it fits to the waves.**

Indeed, thank you for this notion; both are of course driven by solar irradiance in the end. We have reformulated as: "(...) *the motion of the atmosphere is driven by the solar irradiance **and the waves it produces**.*"

**Line 85: There are other efforts of whole atmosphere models: WAM (), GAIA ().**

Thank you for pointing to these models, which have been briefly introduced in the revision.

**Figure 2: Is the depicted neutral wind the total horizontal wind? Similar for the ion drift-is this the ExB drift and is the one perpendicular to the magnetic field?**

We have updated Fig. 2 based on comments made by both Reviewers. In the new version, the depicted neutral winds correspond to the zonal component of the total wind at a high-latitude location (Nordkapp, Norway) near local magnetic midnight, during quiet and storm times. This has been made explicit in the figure caption.

Regarding ion drifts, they were obtained from the momentum equation via post-processing TIE-GCM outputs of the St Patrick storm event. The shown profiles correspond to their zonal component, which has also been made explicit in the caption. One can see that at altitudes above ~150 km the profiles do not exhibit strong variations, as ions are essentially following the ExB drift because collisions with neutrals are scarce (see also the discussion on l. 662 and l. 671).

**Line 199: "total upward energy flux by resolved waves at 100 km" Does this refer to only 100km waves or also larger ones?**

Here, 100 km refers to the altitude where the waves are considered, not their wavelengths. We have clarified it.

**Line 201: "horizontal scales less than 200 km are poorly resolved" Shouldn't waves be resolved with wavelengths approximately 4x the resolution? How does this fit to the 100 km in line 199?**

With the removal of the ambiguity in the previous sentence, this statement no longer seemingly conflicts with it. The fact that horizontal scales below 200 km are poorly resolved is in line with the cited Liu (2016) paper.

**Line 518: "resolution ranges from one orbit to several days" It is not clear to me what is meant here? Orbit averaged to several day averaged? Is this temporal resolution-one measurement pre orbit or every few days?**

Thank you for this comment, which calls for a clarification. This technique intrinsically allows inferring changes in the neutral density based on the orbit tracking of a single object with a temporal resolution of the order of three days or longer (Doornbos et al., 2008). However, by combining orbit data from multiple tracked objects, it is possible to obtain information on neutral density changes at a resolution of 3 h (Storz et al., 2005). We have added this clarification to the revised manuscript.

**Line 661: Maybe the Weimer (2005) empirical ion convection model based on DE-2 data could be mentioned.**

Thank you for suggesting this addition, which we have gladly included in the revised manuscript.

**Line 715: section 3.6 Magnetic fields: I may have missed it but the summary does not mention that the Swarm was able to derive currents without any assumption of current flow due the constellation with nearby satellites. I find this an important point since at the end of the section, the E-region is**

**mentioned and this is the region where strong currents flow. So the interpretation of magnetic fields with respect to current flow without constellation is a challenge (e.g., see the modeling of Maute & Richmond 2017). Maute, A., Richmond, A.D. F-Region Dynamo Simulations at Low and Mid-Latitude. Space Sci. Rev 206, 471–493 (2017).**

Thank you for the suggestion, we added a statement regarding using constellations vs single-spacecraft magnetic field observations, and referred to the Maute and Richmond (2017) paper.

**Line 757: "which are essential also for FACs". It is not quite clear to me what this means? That FAC flows along magnetic field lines?**

We have rephrased the sentence as follows: "*The phenomena are mediated by magnetic field lines, along which FACs flow (Sect. 4.2) and transfer momentum, and whose direction is also essential for the Poynting flux (Sect. 4.3) that transfers energy.*"

**Line 773: "Above the E-layer, electrons and ions drift together and the ionospheric current vanishes." I do not think the authors mean that there is no ionospheric current above the E-region as the sentence suggests. Could this be further explained?**

This is an excellent point; the above statement is misleading and might suggest that there are no currents at all at F-region altitudes. We have rephrased as follows: "*Above the E-layer, electrons and ions **essentially** drift together and the **horizontal** current vanishes.*"

**Technical corrections at the very end ("technical corrections")**

**Line 125: Fig 2. Remind the usual -> reminds of the**

Changed into "recalls".

**Figure 1: I suggest to add the altitude or pressure range of the plots. Does it go from the surface to approximately 500 km?**

Great suggestion, we added in the caption of Fig. 1 that the concentric circles indicate heights of 100, 200 and 500 km.

**Figure 2: It would be easier to add approximate solar local times to the geographic locations of the profiles in the captions.**

Thank you for this suggestion. In the updated version of Fig. 2, we have chosen to rather depict quiet-time vs storm-time profiles at a high-latitude location to illustrate the variability of the parameters. The approximate solar local time corresponding to the profiles was mentioned in the caption, following the suggestion.

**Line 146: reference frames with the neutral gas velocity -> of the neutral gas?**

Implemented.

**Line 212: "In this topic, the" Should this read "in this study/review"?**

We have removed "In this topic", which was redundent with the rest of the sentence.

**Line 300: suggested: to the lower atmosphere**

Implemented.

**Line 564: Should planetary waves be mentioned?**

Added.

**Line 600: Any reason to use speed instead of velocity?**

Corrected.

**Line 648: metre-> meter**

**Line 449: analyser -> analyzer (at least that is the spelling UTD is using)**

Indeed, although we are using British English throughout the manuscript, we agree that for instrument names we should follow their preferred spelling. These have been corrected.

**Line 835: engineering grade magnetometer**

Implemented.

**Line 1083: "propagating tides" upward propagating? Tides can also propagate east-ward and westward. Not all tides reach the F-region.**

Modified as suggested.

**Response to Reviewer #2**

We thank the Reviewer for their very positive and constructive comments on our manuscript. Below we present how we have addressed them in the revision of the paper. The Reviewer's comments are reproduced in bold font, and our responses are given in normal font.

**The article "Lower thermosphere - ionosphere (LTI) quantities: Current status of measuring techniques and models" by M. Palmroth et al. constitutes a large review of the complex physical environment of Earth Lower thermosphere and ionosphere, covering theoretical considerations, experimental techniques and models. The term ignorosphere has been used to indicate this part of the Earth outer space, because of the difficulties of establishing measurement systems that could monitor its physical and chemical parameters on a continuous base globally. Therefore there are still many open scientific questions, that the proposed Daedalus mission could help answering from in-situ measurements. In the Lower thermosphere the forcings from above (principally solar radiation and particle precipitations) deposit large amounts of energy and the forcings from below (principally through tides and waves activity) can significantly affect the physico-chemical processes of both thermosphere and ionosphere. There are such a large varieties of phenomena taking place in this region, that it is not possible to resume all of them in a single review paper, and this paper covers most of them, also pointing out to some phenomena of interest also outside the scientific community, with recently discovered auroral activities. In the article various measuring techniques are described, covering not only the Lower Thermosphere region, but the thermosphere and ionosphere as a whole, with a specific focus to point out the current limitations of this lower altitudinal range and expressing the need for new satellite missions devoted to collect in-situ measurements of the various quantities involved: particles, densities, winds, magnetic fields... Despite the complexity of the topics covered, the article is well organised and refers to recent advances in all the disciplines involved, both from the theoretical and experimental point of view.**

**The article can be accepted for publication.**

**The following remarks are suggestions for minor corrections that could be implemented for improving the overall quality of the manuscript.**

Thank you very much for those encouraging words and for the suggestions!

**Reading the manuscript it clearly appears that it has been written by multiple authors: a few definitions are expressed a couple of times, but without significant overlap.**

Thank you for pointing this out, we have tried to harmonise the style and reduce redundancy in the revision, especially in connection to the next comment below on acronyms and definitions.

**Acronyms are not always defined at first use, sometimes they are defined multiple times e.g. (list non exhaustive): WACCM on page 4, 8, 16 and 19 for WACCM-X, which was already defined on page 4 GPS used on page 19, defined on page 20. GNSS defined on page 39, used on page 19, 20. IPIM defined on page 16 and 19 SuperDARN defined on page 31, used first time on page 23. TomoScand is not defined on page 40.**

Thank you for noting these; we have gone systematically through acronym usage and definitions when preparing the revision. We have also added an appendix with the list of acronyms which appear several times in the manuscript, as suggested by the Editor. Please note that TomoScand is not an acronym, but simply the name of the ionospheric tomography network in Fennoscandia.

**Some figures show ionospheric parameters computed using the International Reference Ionosphere model (IRI). The citation provided and the model shown are not the latest version, which was released in 2016, IRI-2016. The profiles shown might be identical in the IRI-2016, but the latest publications of this model should be cited (e.g., on page 4 and 17): Bilitza, D., D. Altadill, V. Truhlik, V. Shubin, I. Galkin, B. Reinisch, and X. Huang (2017), International Reference Ionosphere 2016: from ionospheric climate to real-time weather predictions, Space Weather, 15, 418–429, doi:10.1002/2016SW001593.**

Thank you for raising this point. We have added the suggested reference to IRI-2016 in the text. However, since Fig. 4 was originally made with the 2012 version, we have kept the figure legend and caption with mentions to IRI-2012.

**I think that captions of figures 1 and 2 should be improved. it is not clear if these two figures show the same altitude range: figure 1 is representing an isobaric surface, while figure 2 a fixed geometrical altitude range. In the text it is indicate that WACCM-X has been extended up to about 500 km, but an explicit indication in the figure caption would be helpful. In the representation of the quantities shown in figure 1, there are no axes to allow the reader to understand where the various panel's quantities are located in altitude.**

Thank you for these suggestions. We have indeed added a mention in the caption of Fig. 1 that the concentric circles indicate altitudes of 100, 200 and 500 km in the panels. To avoid reducing the legibility of the figure, we have not tried to add proper altitude axes in the 3D panels, but hopefully the added information is sufficient to facilitate the interpretation.

Fig. 2 has been updated following suggestions by Reviewer #1, and more information on the profiles, the location they correspond to, as well as the approximate magnetic local time were added in the caption.

Fig. 2 shows profiles from the ground up to 500 km altitude, whereas the topside surface in Fig. 1 is indeed isobaric (corresponding to a pressure value of 4.055e-10 hPa). We do not think adding this value into the manuscript would be meaningful, as we believe the altitude information (concentric circles in each panel) is easier for the reader to interpret within the scope of this review paper.

**On page 11 line 286, the need of a sufficient horizontal resolution and time resolution at low latitudes is expressed, but it is not quantified. I suggest to provide some values that could be used as guidelines for reaching specific scientific goals. This same remark is valid for other conclusions of this article, where no explicit values are indicated.**

This is an excellent point; indeed it would be very valuable to be able to refer to this paper for requirements in the future. We have taken this suggestion into account in the revision and provided quantified (or at least more detailed) needs wherever possible, in each subsection of Sect. 2.

**Figure 4 shows combined results from two different empirical climatological models: NRLMSISE-00 for the neutrals and IRI for the ions. It shows that for some species (N and O) the scale heights of the neutral and its corresponding ion are extremely different. It turns out that in the upper part of the plot the density differences can be of many orders of magnitude. This points out clearly the**

**limitations of these models and an indication that in-situ measurements of both neutrals and ions are necessary.**

Thank you for this comment, we have discussed this point in the revision. Essentially, the reason for the different scale heights for ion and atomic species (e.g., $O^+$ and O) is that there are charge-exchange reactions and other aeronomic processes at play in the upper atmosphere, which affect the corresponding number density profiles differently.

**Page 38, line 1069: following earthquakes, even tsunamis are source of gravity waves observed in the ionosphere (e.g. Makela, J. J., P. Lognonné, H. Hébert, T. Gehrels, L. Rolland, S. Allgeyer, A. Kherani, G. Occhipinti, E. Astafyeva, P. Coïsson, A. Loevenbruck, E. Clévédé, M. C. Kelley, and J. Lamouroux (2011), Imaging and modeling the ionospheric airglow response over Hawaii to the tsunami generated by the Tohoku earthquake of 11 March 2011, Geophys. Res. Lett., 38(13), L00G02).**

Thank you for this suggestion, we have added tsunamis as additional sources of gravity waves (including the provided reference), as well as volcanos and human-made explosions.

**On line 1065 the focus of this part has been put on gravity waves, but some of the phenomena highlighted produce mostly acoustic waves. It is stated correctly that TIDs are observed, which encompass both kind of waves. I think that the unaware reader might not perceive their differences.**

Thank you for pointing this; we have revised this subsection to discuss both acoustic waves and gravity waves. Further, we have also emphasised that investigating the link between the TIDs and gravity waves is one of the objectives for Daedalus.